# Patterning in stratified epithelia depends on cell–cell adhesion

Yosuke Mai[1], Yasuaki Kobayashi[2,3], Hiroyuki Kitahata[4], Takashi Seo[1], Takuma Nohara[1], Sota Itamoto[1], Shoko Mai[1], Junichi Kumamoto[2], Masaharu Nagayama[2], Wataru Nishie[1], Hideyuki Ujiie[1], Ken Natsuga[1]

**Epithelia consist of proliferating and differentiating cells that often display patterned arrangements. However, the mechanism regulating these spatial arrangements remains unclear. Here, we show that cell–cell adhesion dictates multicellular patterning in stratified epithelia. When cultured keratinocytes, a type of epithelial cell in the skin, are subjected to starvation, they spontaneously develop a pattern characterized by areas of high and low cell density. Pharmacological and knockout experiments show that adherens junctions are essential for patterning, whereas the mathematical model that only considers local cell–cell adhesion as a source of attractive interactions can form regions with high/low cell density. This phenomenon, called cell–cell adhesion-induced patterning (CAIP), influences cell differentiation and proliferation through Yes-associated protein modulation. Starvation, which induces CAIP, enhances the stratification of the epithelia. These findings highlight the intrinsic self-organizing property of epithelial cells.**

## Introduction

Epithelial tissue is composed of layers of cells that cover the surfaces of organs, and their homeostasis is maintained through the spatial organization of cells with various fates. Epithelial cells are distributed in a coordinated manner and typically exhibit visible patterns in stratified epithelia, such as human fingerprints ([1]), bird feathers ([2]), and mouse tail scales ([3], [4], [5]), at the macroscopic level. Genetic, developmental, and environmental factors contribute to the unique arrangement of stem cells, proliferating cells, and differentiating cells in tissues, leading to context-dependent epithelial patterning.

Keratinocytes are the primary cells in the epidermis, the stratified epithelium of the skin. Previous studies have highlighted keratinocytes' ability to develop patterns. Seminal research by Green and Thomas demonstrated that human keratinocytes formed patterns resembling human fingerprints when cultured on feeder cells ([6]). Subsequent studies revealed that human keratinocytes can self-organize into clusters expressing stem cell markers ([7], [8]), implying that the multicellular arrangements might affect cell fate. However, the characteristics and mechanisms underlying keratinocyte patterning are not yet thoroughly understood.

Cell–cell adhesion is a characteristic of epithelia and is supported by specialized junctional complexes, including adherens junctions (AJs). AJs are composed of cadherins and catenins and provide mechanical attachments between neighboring epithelial cells ([9]). Perinatal lethality has been observed in mice lacking any of the AJ components ([10], [11], [12], [13]), underscoring the significance of AJs.

Here, we show that cell–cell adhesion governs keratinocyte patterning. The patterning is mediated by AJs and facilitates cell dynamics with the help of the Yes-associated protein (YAP) pathway. Our findings elucidate the molecular and cellular basis underlying the spatial organization of cells in the epidermis.

## Results

### Spontaneous patterning of keratinocytes

First, we observed the morphology of epithelial cell sheets by using HaCaT cells, an immortalized keratinocyte line ([14]). The cells were seeded in a high-calcium medium containing 10% FBS and allowed to reach full confluence in 1 d (day 1; Fig 1A). On day 4, the cells displayed a self-organized pattern, comprising areas of high and low cell density (Fig 1A and B). HaCaT cells were originally derived from the back skin of a 62-yr-old male and not from a single cell ([14]); therefore, heterogeneous cells might have formed the regions of high or low cell densities. To rule out this possibility, we performed single-cell cloning of

[1]Department of Dermatology, Faculty of Medicine and Graduate School of Medicine, Hokkaido University, Sapporo, Japan  [2]Research Center of Mathematics for Social Creativity, Research Institute for Electronic Science, Hokkaido University, Sapporo, Japan  [3]Department of Mathematics, Faculty of Science, Josai University, Sakado, Japan  [4]Department of Physics, Graduate School of Science, Chiba University, Chiba, Japan

Correspondence: yosukemaiderma@gmail.com; natsuga@med.hokudai.ac.jp

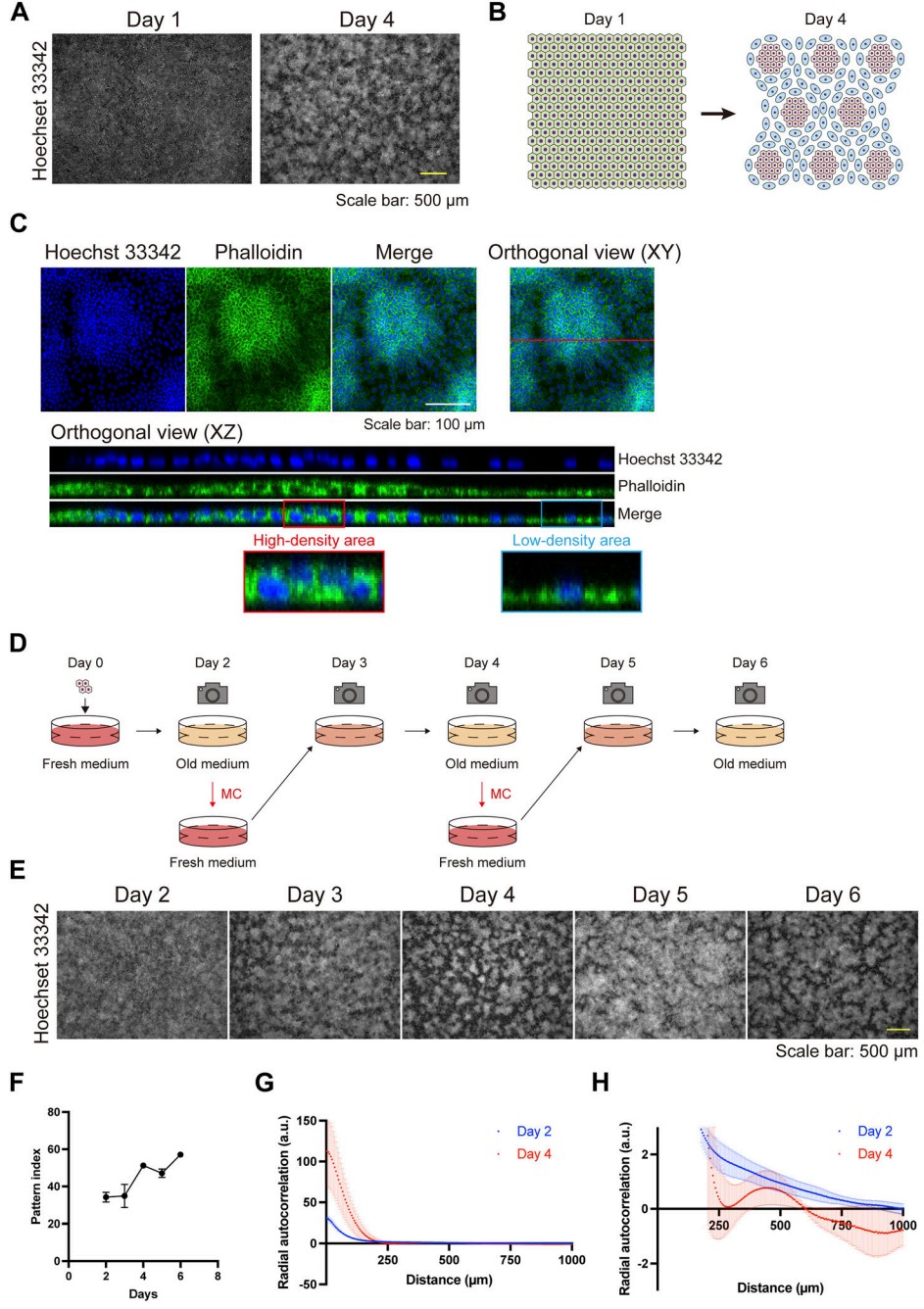

**Figure 1.  The self-organizing pattern of keratinocytes.**
**(A)** Immunofluorescent images of keratinocytes displaying a self-organized pattern 1 and 4 d after seeding. Nuclei are labeled with Hoechst 33342. Scale bar: 500 μm. **(B)** Schematic diagram of keratinocyte pattern, comprising areas of high and low cell density. **(C)** Orthogonal immunofluorescent images of keratinocyte pattern. Blue indicates nuclear labeling with Hoechst 33342. Green indicates actin labeling with phalloidin. The XZ orthogonal images correspond to the red line of the XY orthogonal view. The red and blue rectangles are enlarged to show a high-density area and a low-density area, respectively. Scale bar: 100 μm. **(D)** Schematic diagram of medium changes (MCs) and observation points. **(E)** Immunofluorescent images of keratinocytes following MCs. Nuclei are labeled with Hoechst 33342. Scale bar: 500 μm. **(F)** Pattern index during the cell culture. Data are presented as mean values ± SEM. N = 16 for each time point. **(G)** The radial autocorrelation function of fluorescent images of cultured keratinocytes 2 and 4 d after seeding. Data are presented as mean values ± SEM. N = 3 for each time point. **(G, H)** Enlarged view of (G).

HaCaT cells by integrating a puromycin cassette and the *Cas9* gene into the cells, the latter of which enabled further knockout experiments (15). Single-cell-cloned HaCaT cells developed a self-organized pattern similar to that observed in the parental cells (Fig S1), indicating the presence of an intrinsic cell property that gives rise to regions of high or low cell density. Morphologically, phalloidin staining revealed that cells in the areas of high cell density were cuboidal, compact, and stratified, whereas those in the areas of low cell density were flat (Figs 1C and S2).

Time-lapse images of keratinocytes during patterning revealed that cells initially moved randomly, and subsequently, the areas of high and low cell density developed spontaneously (Fig S3A and B and Video 1). This self-assembling pattern was obscured 1 d after a medium change (MC) (day 4 to day 5; Fig 1D and E) and reappeared on day 6 (Fig 1D and E). Time-lapse images showed that cells in the areas of high cell density migrated toward regions of low cell density after MCs and that the areas of high/low cell density re-formed (Fig S4A–D; Video 2 and Video 3). The pattern was quantified with an ImageJ plugin, in which the pattern index increases as the

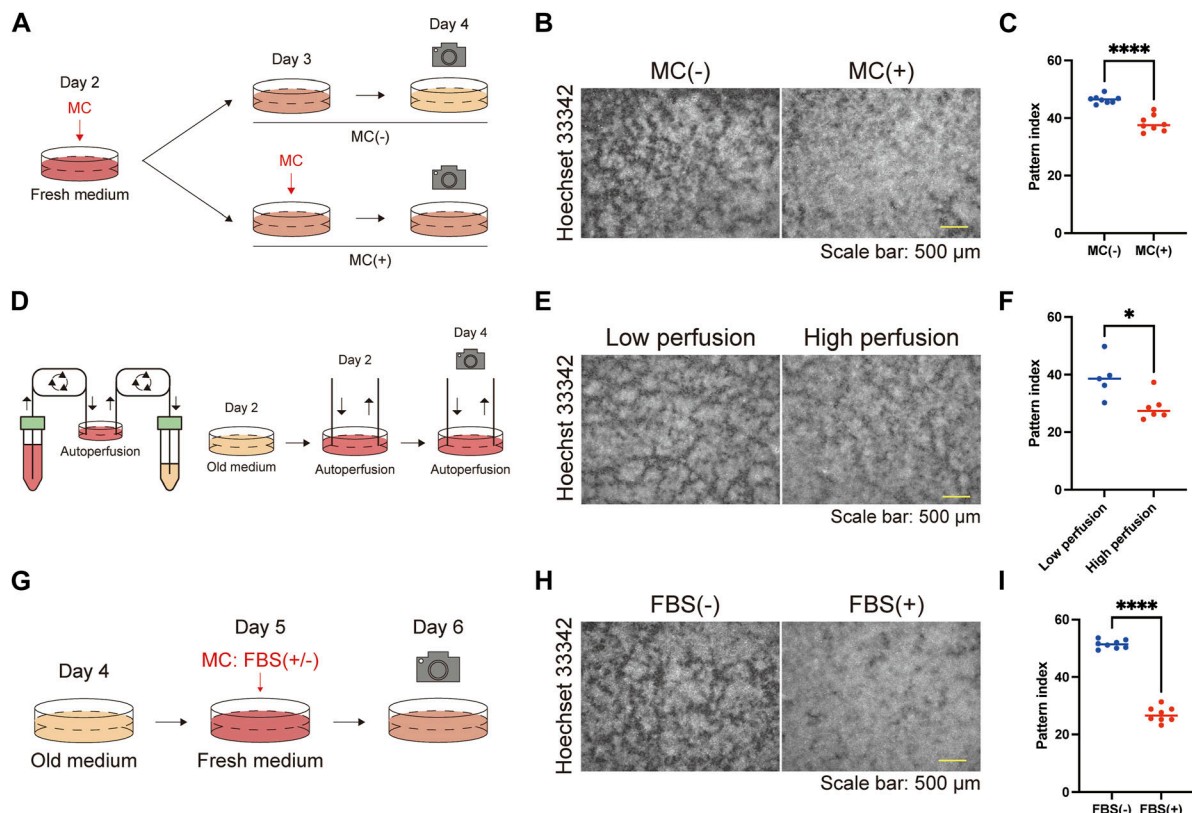

**Figure 2. Serum starvation induces keratinocyte pattern formation.**
**(A)** Schematic diagram of the experiments comparing keratinocyte pattern formation with or without MC. **(B)** Immunofluorescent images of keratinocytes with or without MC. Nuclei are labeled with Hoechst 33342. Scale bar: 500 $\mu m$. **(C)** Pattern index with or without MC. N = 8 for each group. **(D)** Schematic diagram of the autoperfusion culture system and experiment. **(E)** Immunofluorescent images of keratinocytes cultured under low and high perfusion rates on day 4. Nuclei are labeled with Hoechst 33342. Scale bar: 500 $\mu m$. **(F)** Pattern index for cultures with low and high perfusion rates. N = 5 for the culture group with the low perfusion rate and N = 6 for the culture group with the high perfusion rate. All data are presented as mean values and analyzed with two-tailed Mann–Whitney $U$ tests. *$P < 0.05$. **(G)** Schematic diagram of the experiments comparing keratinocyte patterns after MC with or without FBS. **(H)** Immunofluorescent images of keratinocytes after MC with or without FBS. Nuclei are labeled with Hoechst 33342. Scale bar: 500 $\mu m$. **(I)** Pattern index after MC with or without FBS. All data are presented as mean values. Statistical analysis was performed with two-tailed Mann–Whitney $U$ tests. *$P < 0.05$. ****$P < 0.0001$.

distinction between areas of high and low cell density becomes more evident (see the Materials and Methods section). This quantitative analysis confirmed that the pattern was disrupted by MCs (Fig 1F). To further confirm that the areas of high and low cell density exhibited a pattern—defined as regularly repeated cell arrangement—and were not merely an inhomogeneous distribution of cell densities, the images of the cells at day 2 (patternless) and day 4 (pattened) were analyzed using autocorrelation functions (Figs 1G and H and S5). On day 2, the radial autocorrelation gradually decreased with distance, indicating that no pattern was present. In contrast, on day 4, the autocorrelation showed a pronounced first-ordered nadir corresponding to the average nearest distance between high cell density areas, demonstrating the presence of patterning (Fig 1G and H).

## Serum starvation induces the self-organizing pattern of keratinocytes

The time course studies (Figs 1D–F and S3A and B and S4A–D; Video 1 and Video 2) led us to speculate that the starvation of the culture

medium induced the formation of keratinocyte patterns. To test this hypothesis, we compared groups with and without MCs and found that the pattern was disturbed in the group with MCs (Fig 2A–C). The automated perfusion culture system (Fig 2D) demonstrated that the procedure of the MC itself did not significantly impact the phenotype, as the cells maintained at a low perfusion rate exhibited more pronounced patterns than those maintained at a high perfusion rate (Fig 2D–F). To further investigate the factors contributing to pattern formation in the culture medium, we compared cells that underwent MCs with or without 10% FBS (Fig 2G–I). 1 d after the MC, the pattern disappeared in the FBS group but not in cells without FBS (Figs 2G–I and S6A and B; Video 4). These results indicate that serum starvation is crucial for keratinocyte patterning.

## Cell–cell adhesion through adherens junctions is essential for pattern development

To elucidate the underlying mechanism(s) behind the pattern formation, we performed RNA sequencing (RNA-seq) analysis to

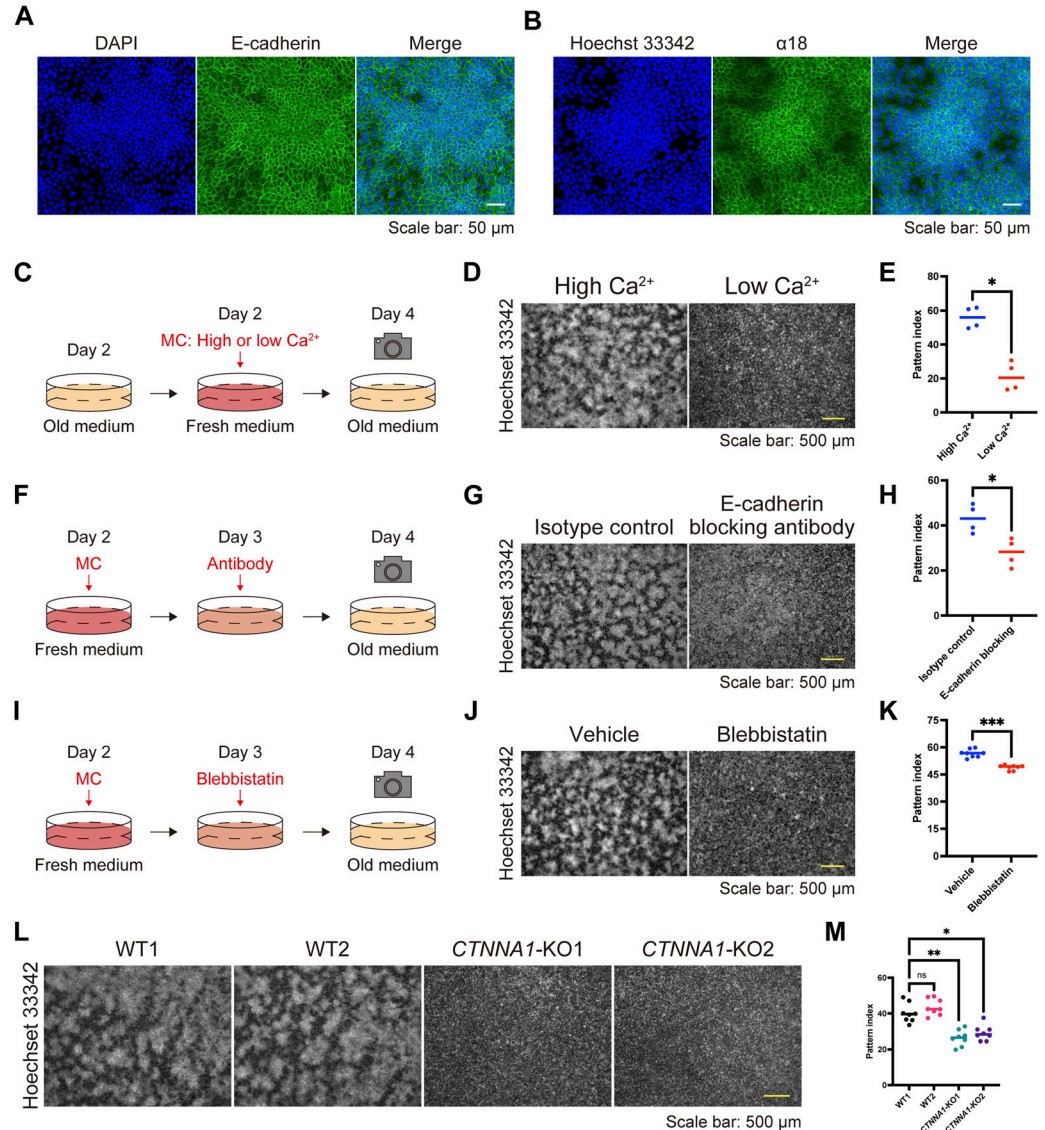

**Figure 3. Adherens junctions regulate keratinocyte patterning.**
**(A, B)** Immunofluorescent images of keratinocytes on day 4. Nuclei are stained with DAPI and Hoechst 33342 (blue). **(A, B)** Cell–cell adhesions are visualized with E-cadherin (A) and α18 labeling (B) (green). Scale bar: 50 $\mu$m. **(C)** Schematic diagram of experiments investigating patterns under high-calcium (1.8 mM) and low-calcium (0.06 mM) conditions. **(D)** Immunofluorescent images of keratinocytes under high- and low-calcium conditions. Nuclei are labeled with Hoechst 33342. Scale bar: 500 $\mu$m. **(E)** Pattern index in high- and low-calcium conditions. N = 4 for each group. **(F)** Schematic diagram of experiments investigating patterns with or without the E-cadherin-blocking antibody. **(G)** Immunofluorescent images of keratinocytes with or without the E-cadherin-blocking antibody. Nuclei are labeled with Hoechst 33342. Scale bar: 500 $\mu$m. **(H)** Pattern index with or without the E-cadherin-blocking antibody. N = 4 for each group. **(I)** Schematic diagram of experiments investigating patterns with or without blebbistatin, a non-muscle myosin II inhibitor. **(J)** Immunofluorescent images of keratinocytes with or without blebbistatin. Nuclei are labeled with Hoechst 33342. Scale bar: 500 $\mu$m. **(K)** Pattern index with or without blebbistatin. N = 8 for each group. **(L)** Immunofluorescent images of WT and *CTNNA1*-knockout (KO) keratinocytes. Nuclei are labeled with Hoechst 33342. Scale bar: 500 $\mu$m. **(M)** Pattern index of WT and *CTNNA1*-KO keratinocytes. N = 8 for each group. All data are presented as mean values. Data for (E, H, K) were analyzed with two-tailed Mann–Whitney $U$ tests. Data for (M) were analyzed with the Kruskal–Wallis test followed by Dunn's multiple comparison test. ns, not significant. *$P < 0.05$; **$P < 0.01$; ***$P < 0.001$.

compare HaCaT cells cultured under high- or low-density conditions, as the pattern consisted of areas of high and low cell density. The gene ontology biological process terms enriched among the differentially expressed genes through DESeq2 (16) included cell adhesion and keratinocyte differentiation (Fig S7A–C and Table S1). The RNA-seq data prompted us to examine the distribution of AJ molecules, such as E-cadherin and actin, given that AJs are main contributors to cell adhesion. We found that these molecules were localized at intercellular junctions in areas of high cell density (Figs 1C and 3A and S8A–C). E-cadherin and α-catenin form an AJ complex (9), and through AJ development, α-catenin undergoes a conformational change because of intercellular forces, which is recognized by the α18 antibody (17). α18 labeling was also pronounced in areas of high cell density (Figs 3B and S8D–H). These data suggest that cells in regions of high cell density form AJs in response to intercellular forces.

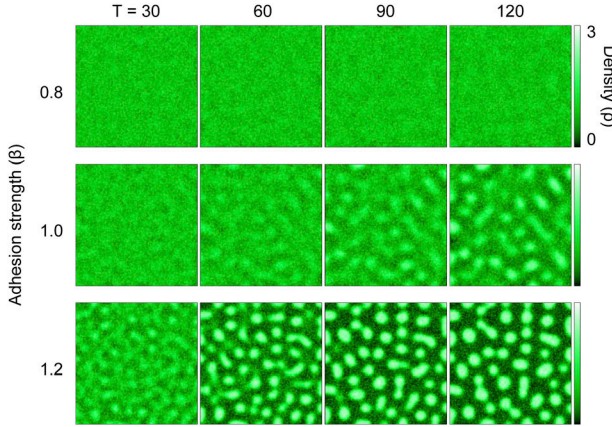

**Figure 4. A mathematical model simulates a spatial density pattern using two variables: cell density and stress caused by adhesion.**
A two-dimensional cell density model predicted the emergence of spatial patterns of cell distribution, depending on adhesion strength. The color represents the local cell density $\rho$. Simulations for three different values of $\beta$ (adhesion strength) are shown, with time moments T [a.u.] = 30, 60, 90, and 120.

To investigate the role of AJs in pattern development, we first compared cells under low- and high-calcium conditions because calcium is necessary to form AJs (18). A high/low cell density pattern developed under high-calcium conditions but was not observed under low-calcium conditions (Fig 3C–E). Similarly, treatment with SHE78-7 antibody, which blocks E-cadherin-mediated cell adhesion (19, 20), inhibited patterning (Fig 3F–H). The force-dependent conformational change of $\alpha$-catenin is induced by myosin-II (17), and its inhibition by (–)-blebbistatin (21, 22) also disturbed pattern formation (Fig 3I–K). *CTNNA1*-knockout (KO) HaCaT cell lines, in which $\alpha$-catenin expression was nullified by gene editing (Fig S9A and B), exhibited the same phenotype (Fig 3L and M). These results demonstrate that AJs are essential for forming keratinocyte patterns.

### Cell–cell adhesion can contribute to pattern development

We then asked whether cell–cell adhesion is sufficient for the formation of keratinocyte patterns and used mathematical modeling to answer this question. We used a two-dimensional continuous model consisting of two variables: cell density and stress caused by adhesion. The main assumptions of the model were that the collective movement of cells is driven by the spatial imbalance of adhesion strength and that cell–cell adhesion increases with cell density. Note that these assumptions themselves do not tell whether density patterning will emerge. We performed simulations by varying the coefficient of adhesion strength as a control parameter. Each simulation began with spatially uniform density and stress. We observed that for sufficiently strong adhesion, the initial uniform distribution became unstable, and a spatial pattern of density emerged over time. By contrast, for weak adhesion, no coherent pattern appeared (Fig 4; Video 5, Video 6, and Video 7). These patterns were robust against noise strengths accounting for the random movements of individual cells (Fig S10). These results suggest that adequately strong cell–cell adhesion can contribute to the emergence of density patterns.

### Keratinocyte patterns spatially dictate cell proliferation and differentiation

We further characterized keratinocyte differentiation through our experiments, as indicated by the RNA-seq data (Fig S7B and C). Keratin 10 (KRT10)-positive differentiated cells were abundant and stratified in areas of high cell density (Fig 5A and B). By contrast, phospho (Thr3)-monomethyl (Lys4) histone H3 (PH3)-positive proliferative cells were found in areas of low cell density (Fig 5C). These data suggest that differentiation and proliferation are spatially regulated through pattern formation. Because AJs are essential for forming the high/low cell density pattern, we wondered whether they were also involved in regulating pattern-dependent differentiation and proliferation. *CTNNA1*-KO HaCaT cells failed to show pattern-dependent differentiation (Fig 5D and E), suggesting that AJs involving $\alpha$-catenin play a role in this process.

We then focused on the YAP pathway, as the coordinated pattern-dependent differentiation and proliferation seemed to be linked to cell density, and YAP, a "crowd control" molecule, uses $\alpha$-catenin to sense cell density in keratinocytes (23). The localization of YAP, which is regulated by cell density, determines cell fate; cytoplasmic YAP induces differentiation, and nuclear YAP promotes proliferation (23, 24, 25, 26). As expected, cytoplasmic YAP and nuclear YAP were found in areas of high and low cell density, respectively (Fig 6A and B). In line with YAP dynamics, ANKRD1, a YAP downstream molecule (25, 27, 28), was localized to the nuclei of cells in high-density areas but not in low-density areas (Fig 6C and D). The YAP activator PY-60, which induced nuclear localization of YAP and ANKRD1 in keratinocytes of high cell density areas (Fig S11A–D), inhibited cell differentiation (Fig 6E and F) and disrupted pattern formation (Fig S11E and F). This disturbed patterning might have resulted from the impaired contact inhibition of proliferation (29) or the dysregulation of AJs by YAP activation (30). By contrast, YAP inhibition by a tankyrase inhibitor XAV939, which prevented the nuclear localization of YAP and ANKRD1 in keratinocytes of low cell density areas (Fig S11G–J), suppressed cell proliferation (Fig 6G and H) and slightly modified pattern formation (Fig S11K and L). These results suggest that YAP modulates the differentiation and proliferation states of patterned keratinocytes.

### Serum starvation contributes to epidermal stratification

Finally, we asked whether the keratinocyte patterns induced by serum starvation impacted the stratification of the epidermis, as cells in areas of high cell density within the pattern were predisposed to differentiation. Air-liquid interface culture of HaCaT cells, which induces stratification, revealed that the epidermis was thicker in serum-starved conditions than in serum-rich conditions, in which pattern formation was impaired (Figs 7A and B and S12A and B). As expected, KRT10-positive layers in serum-starved conditions were thicker than those in serum-rich conditions, whereas the thickness of KRT14-positive layers was comparable between the two groups (Fig S12C–E). In addition, *CTNNA1*-KO HaCaT cells failed to form a stratified epithelium (Fig 7C).

We further evaluated whether the serum-starved condition could affect epidermal stratification ex vivo. We used a wound-healing mouse model with a suction blister to observe re-

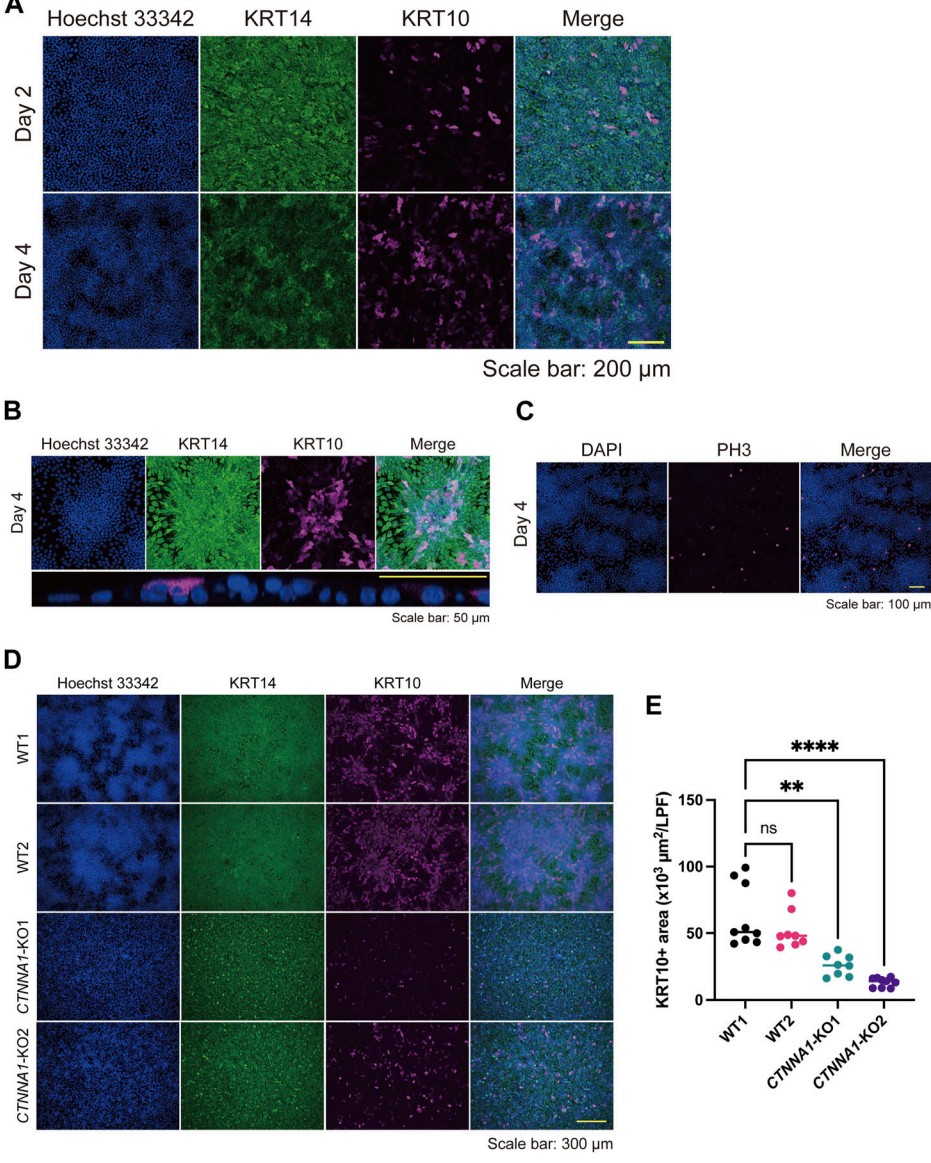

**Figure 5. Patterning correlates with differentiation and proliferation markers in keratinocytes.**
**(A)** Immunofluorescent images of keratinocytes on days 2 and 4. Hoechst 33342 (blue); Keratin (KRT) 14, a basal keratinocyte marker (green); KRT10, a differentiated keratinocyte marker (magenta). Scale bar: 200 $\mu$m. **(B)** The upper panels are immunofluorescent images of a high-density region on day 4. The lower panel is an orthogonal YZ image of a high-density region under high magnification. Hoechst 33342 (blue); KRT14 (green); KRT10 (magenta). Scale bar: 50 $\mu$m. **(C)** Immunofluorescent images of keratinocytes on day 4. DAPI (blue). PH3, a proliferation marker (magenta). Scale bar: 100 $\mu$m. **(D)** Immunofluorescent images of WT and *CTNNA1*-KO keratinocytes on day 4. Hoechst 33342 (blue); KRT14 (green); KRT10 (magenta). Scale bar: 300 $\mu$m. **(E)** KRT10-positive (KRT10+) areas per low power field of WT and *CTNNA1*-KO keratinocytes per low power field. N = 9 for WT1 and *CTNNA1*-KO2. N = 8 for WT2 and *CTNNA1*-KO1. All data are presented as mean values. Data for (E) were analyzed with the Kruskal–Wallis test followed by Dunn's multiple comparison test. **P < 0.01. ****P < 0.0001.

stratification after wounding (31). A sample of mouse back skin with a suction blister wound was cut out and cultured in serum-starved and serum-rich conditions (Fig 7D). Re-epithelialization was completed on day 1, and re-stratification began. On day 4, the re-stratified epidermis cultured in serum-starved conditions was thicker than that cultured in serum-rich conditions (Fig 7E and F). These data imply that appropriate serum starvation facilitates epidermal stratification through cell–cell adhesion and the subsequent patterning of the cells.

## Discussion

Our study presents a novel and robust model of cell–cell adhesion-induced patterning (CAIP). CAIP is mediated by AJs and spatially regulates the differentiation and proliferation of epithelial cells.

During skin development, such as placode formation, the periodic expression of signaling molecules is thought to arise in the early stages of development, dictate cell behaviors, and orchestrate follicular patterning in the skin (32, 33, 34). However, recent studies have revealed that periodic follicle patterning is triggered by mechanical rather than molecular events (2, 35). In this scenario, follicular patterns can arise from mechanical instability caused by fibroblast contraction. Then, self-organized fibroblast aggregation through contractility-driven cellular pulling triggers the mechanosensitive activation of $\beta$-catenin in neighboring keratinocytes, activating the follicle gene expression program. These studies indicate that cellular contraction mechanics in the mesenchyme could organize epithelial patterning and subsequent cell fate decisions in vivo. In contrast to the follicle placodes, fingerprint ridge formation, which involves sweat gland development (1), is facilitated by signaling molecules, including EDAR, WNT, and BMP,

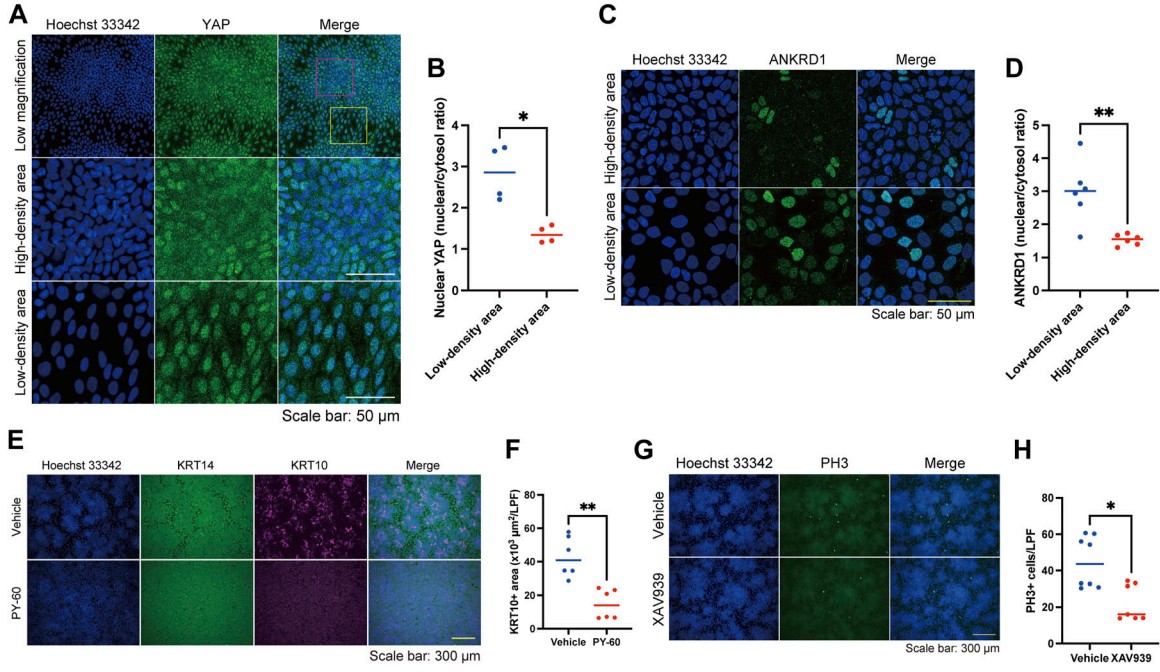

**Figure 6. Yes-associated protein (YAP) modulates the differentiation and proliferation states of patterned keratinocytes.**
**(A)** Immunofluorescent images of keratinocytes on day 4. Hoechst 33342 (blue). YAP, a cell density-sensitive molecule (green). The red and yellow squares indicate a dense area and a sparse area, respectively. Scale bar: 50 µm. **(B)** Comparison of the nuclear YAP intensity between keratinocytes in low and high cell density areas. The nuclear YAP intensity was quantified using the nuclear and cytosol intensity ratio. N = 6 for each group. **(C)** Immunofluorescent images of patterned keratinocytes in a high- or a low-density area. Hoechst 33342 (blue). ANKRD1, a YAP downstream molecule (green). Scale bar: 50 µm. **(D)** Comparison of the nuclear ANKRD1 intensity between keratinocytes in high and low cell density areas. The nuclear ANKRD1 intensity was quantified using the nuclear and cytosol intensity ratio. N = 6 for each group. **(E)** Immunofluorescent images of keratinocytes with or without PY-60, a YAP activator. Hoechst 33342 (blue); KRT14 (green); KRT10 (magenta). Scale bar: 300 µm. **(F)** KRT10+ areas per low power field of immunofluorescent images for keratinocytes with or without PY-60. N = 6 for each group. **(G)** Immunofluorescent images of keratinocytes with or without XAV939, a tankyrase inhibitor. Hoechst 33342 (blue); PH3 (green). Scale bar: 300 µm. **(H)** Numbers of PH3-positive (PH3+) cells with or without XAV939. N = 8 for each group. All data are presented as mean values. Data for (B, D, F, H) were analyzed with two-tailed Mann–Whitney U tests. ns, not significant. *P < 0.05. **P < 0.01.

and does not recruit mesenchymal cells. The concept of CAIP is distinct because epithelial cell–cell adhesion is essential and sufficient for the patterning. Research to determine whether CAIP is involved in other in vivo epithelial patterning is warranted.

CAIP also imitates the periodic buckling of the human epidermis, in which epidermal stem cells with high β1 integrin expression are localized in the areas facing dermal protrusion and transient amplifying cells are present in the epidermal rete ridge structures (36, 37). Keratinocytes cultured on undulating elastomer substrates resembling rete ridge structures display a similar stem cell distribution, in which cells with high β1 integrin expression cluster at the tips of the topographies (38, 39). The disruption of cell–cell adhesions by Rho kinase inhibitors impairs confluent sheet formation on undulating substrates. However, the stem cell clusters on undulating substrates strongly express E-cadherin and F-actin, in contrast to the high E-cadherin expression in the differentiation region of CAIP. This discrepancy might be explained by differences in the experimental designs.

CAIP aligns with the principles of mechanobiological cell fate decisions in epithelial sheets. In a confluent of monolayer of MDCK cells, topological defects govern α-catenin and YAP-associated cell death and extrusion (40, 41). Interestingly, HaCaT cell monolayers display a distinct phenotype from MDCK monolayers; they are more elatic (42) and have a characteristic of the multi-stratified

organization. Miroshnikova et al demonstrated that in keratinocytes, differentiation occurs near proliferating cells, resulting from cell shape distortion, increased cell–cell adhesion, and decreased cortical tension in a confluent state (43). In our CAIP model, serum starvation induces cell gathering, which may distort cell shape and increase cell–cell adhesion, affecting cortical tension and triggering differentiation. Furthermore, CAIP may also be related to the mechanosensitive ERK pathway because cell deformation couples with ERK activation, and ERK activities regulate keratinocyte differentiation (44, 45).

For the mechanobiological process in keratinocytes, AJs are essential for intercellular junction formation (46) and serve as a major hub for Hippo-YAP pathway components such as Merlin (NF2), angiomotin (AMOT), and annexin A2 (ANXA2) (29, 47, 48, 49). AMOT, a Hippo pathway regulator, is localized at AJs and binds to YAP, leading to the translocation of YAP from the nucleus into the cytoplasm (50, 51, 52). To regulate AMOT activity, tankyrases associate with AMOT and promote AMOT degradation (53, 54). We showed that a tankyrase inhibitor, XAV939, impairs the nuclear localization of YAP and ANKRD1 (Fig S11G–J). These findings suggest that pattern-dependent YAP activation may be regulated by tankyrase-AMOT-YAP interaction. Furthermore, ANXA2 directly associates with YAP at AJs in response to increased cell density (29, 55). Treatment with PY-60 releases the ANXA2-YAP complex from the

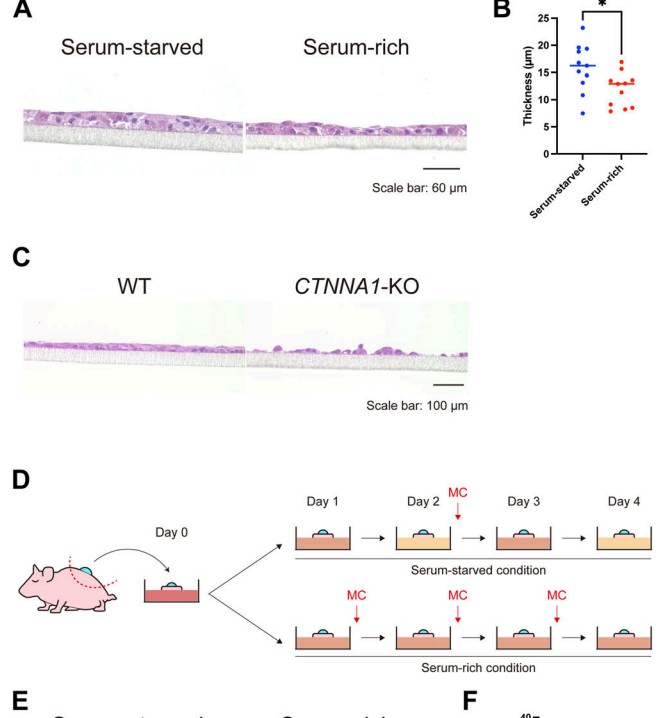

**Figure 7. Air-liquid interface in vitro and ex vivo cultures under serum-starved and serum-rich conditions.**
**(A)** Hematoxylin and eosin (H&E) staining of air-liquid interface HaCaT keratinocyte cultures in serum-starved and serum-rich conditions. Scale bar: 60 μm. **(B)** Quantification of epidermal thickness in the air-liquid interface cultures under serum-starved and serum-rich conditions. N = 11 for each group. **(C)** H&E staining of WT and *CTNNA1*-KO air-liquid interface cultures. Scale bar: 100 μm. **(D)** Schematic diagram of ex vivo culture experiment using P1 neonate back skin with suction blister wound under serum-starved and serum-rich conditions. **(E)** H&E staining of ex vivo cultured tissue. Scale bar: 60 μm. **(F)** Quantified re-stratified epidermal thickness in the ex vivo culture. N = 6 for each group. All data are presented as mean values and were analyzed with two-tailed Mann–Whitney *U* tests. *P < 0.05.

cell membrane, activating YAP-driven transcription (29). In CAIP, PY-60 treatment induced nuclear localization of YAP and ANKRD1 in keratinocytes of high cell density areas (Fig S11A–D), indicating that ANXA2 binds to YAP and inhibits its transcriptional activities. Of note, XAV939 and PY-60 also altered the keratinocyte patterning itself (Fig S11E, F, K, and L), which may be explained by the fact that AJ-associated Hippo-YAP pathway components such as NF2 and ANXA2 are required for the formation of AJs (55, 56). As the Hippo-YAP pathway is a key regulator for epidermal stratification (23, 57, 58), our CAIP model may deepen our understanding of the Hippo-YAP pathway in epidermal stratification.

Serum starvation induces quiescence in vitro (59) and has been used for cultured keratinocytes (60, 61, 62), fibroblasts (63, 64), and other cell types (65, 66, 67). By contrast, serum stimulation

enhances keratinocyte migration (60, 68, 69). Hence, CAIP in serum-starved conditions might reflect the resting state of the epidermis. However, in our study, CAIP and serum starvation also supported epithelial stratification, which could simulate epidermal morphogenesis or the final stage of wound healing in the epidermis. These data indicate that CAIP modulation might help finalize wound closure by enhancing epidermal stratification once the wound gap is filled with one layer of epithelial cells. In line with this, evidence indicates that hyperhydration or excessive extracellular fluid delays wound closure in clinical settings (70, 71). Theoretically, the serum-starved culture, which reduces the MC frequency, could be beneficial to other epidermis or skin organoid cultures to obtain thicker epidermal sheets efficiently and economically. Our findings may also have applications in regenerative medicine, such as the preparation of epidermal grafts.

As the mathematical modeling indicated patterning without the need for stratification steps, it is possible that CAIP is not a direct consequence of keratinocyte stratification. However, our experimental approach did not distinguish whether CAIP occurred before, or concurrently with, cell stratification, which represents a limitation of our study. The model's failure to incorporate cell stratification adds another limitation. Moreover, our study was not able to identify what molecules in the serum were responsible for suppressing CAIP because the serum is a complex mixture of biomolecules, including hormones, growth factors, vitamins, and other nutrients. Further research is required to address these issues.

In conclusion, our study uncovered the pivotal role of cell–cell adhesion in modulating epithelial cell patterning. Our CAIP model deepens our mechanistic insight into cellular organization and its consequences for cell fate decisions and epithelial stratification.

# Materials and Methods

## Cells

HaCaT cells (14) were originally obtained from Dr. Norbert Fussenig's lab (German Cancer Research Center), and the cell identity was confirmed with the Cell Culture STR profile (Biologica). Single-cell-cloned HaCaT cells with stable Cas9 expression were established previously (15). Briefly, HaCaT cells were transfected with pLenti-EF1a-Cas9-Puro lentiviral particles (Applied Biological Materials) and selected with 1 μg/ml puromycin (Thermo Fisher Scientific). Thereafter, 50 cells were seeded into a 10 cm dish, and the single-cell clone was obtained with Scienceware cloning discs (Sigma-Aldrich) dipped in trypsin (Wako). To establish *CTNNA1* HaCaT cells, crRNA was acquired from predesigned Alt-R CRISPR-Cas9 guide RNA (Integrated DNA Technologies); the crRNA contained the TGAAGC-GAGGCAACATGGTT sequence with the CGG PAM sequence targeting exon 4 of the *CTNNA1* genomic sequence (NG_047029.1). For transfection, HaCaT cells stably expressing Cas9 were seeded onto a 24-well plate with a concentration of 300 × 10^5 cells/well. 1 d after seeding, HaCaT cells were transfected with a 10 nM duplex of crRNA and Alt-R CRISPR-Cas9 tracrRNA tagged with ATTO 550 (Integrated DNA Technologies) using Lipofectamine RNAiMAX Transfection

Reagent (Thermo Fisher Scientific) according to the manufacturer's instructions. Thereafter, the single-cell clone was obtained by seeding onto a 96-well plate. The truncations on the *CTNNA1* gene caused by CRISPR/Cas9 were confirmed by Sanger sequencing, and the knockout of the *CTNNA1* gene was examined by Western blot. All cell stocks were routinely tested for mycoplasma contamination, and all tests were negative.

## Cell culture

HaCaT cells were maintained in DMEM with 4.5 g/liter glucose and L-glutamine (Nacalai Tesque) supplemented with 10% FBS (HyClone) and 1× antibiotic–antimycotic mixed stock solution (Nacalai Tesque) at 37°C in humidified air with 5% $CO_2$. For the live imaging with confocal microscopy, 4.5 g/liter glucose DMEM without phenol red (Nacalai Tesque) supplemented with L-glutamine, 10% FBS, and 1x antibiotic–antimycotic mixed stock solution was used for cell culture. Low-calcium-condition experiments used calcium-free 4.5 g/liter glucose DMEM with or without phenol red (Nacalai Tesque) supplemented with L-glutamine, 10% calcium-free FBS, and 1× antibiotic-antimycotic mixed stock solution. To prepare calcium-free FBS, 5 g Chelex 100 (Bio-Rad) was added to 100 ml FBS with shaking for 1 h at RT, followed by filtration with a 0.22 $\mu$m Stericup-GP Express Plus PES (Millipore).

## Pattern formation of cultured keratinocytes

HaCaT cells were seeded at $5.0 \times 10^5$ cells/ml onto $\mu$-Slide 8 Well Chambers (ibidi) using 250 $\mu$l of suspension per well or onto a $\mu$-Dish 35 mm dish (ibidi) using 2 ml of suspension per dish. The culture medium was replenished every 2 d with equivalent volumes unless otherwise specified. The keratinocyte pattern was observed 4 d after seeding.

## Inhibitor treatments

HaCaT cells were treated with PY-60 (Axon Medchem) at a concentration of 10 $\mu$M when the culture medium was replenished 2 d after seeding. The cells were observed 2 d after treatment. E-cadherin-blocking antibody (SHE78-7; Takara), blebbistatin (Cayman Chemical), or XAV939 (Fujifilm) were used at a concentration of 30 $\mu$g/ml, 12.5, or 3 $\mu$M, respectively, 3 d after seeding. The cells were evaluated 4 d after seeding.

## Air-liquid interface epidermal culture

ThinCert 12-well cell culture inserts with 0.4 $\mu$m pores (Greiner Bio-One) were placed on Falcon 12-well cell culture plates (Corning) and precoated with CTS CELLStart Substrate (Thermo Fisher Scientific) in a 1:50 dilution of Dulbecco's phosphate-buffered saline (PBS) with $MgCl_2$ and $CaCl_2$ (Sigma-Aldrich) overnight. Subsequently, 1 ml of culture media was added into the well, and 1 ml of HaCaT cell suspension at a concentration of $2.5 \times 10^5$ cells/ml was seeded onto the inserts. In the serum-starved condition, the culture medium for the insert and the well was replenished 2 d after seeding. In the serum-rich condition, the medium was replenished every day. For

placing the inserts under an air-liquid interface culture, Extra Thick Blot Filter Paper (Bio-Rad) was cut to 2.8 cm × 2.8 cm with two 12 mm holes. The filter paper was placed into a well of Falcon six-well deep well plates (Corning), which was filled with 11.5 ml of the cell culture medium. 4 d after seeding, the culture medium within the insert was removed in both conditions, and the inserts were placed on the filter papers. The culture medium in the well was replenished every 3 d for the serum-starved condition and every day for the serum-rich condition for 14 d after seeding. Samples were collected by excising the membranes from the inserts and were fixed with formalin.

## Ex vivo skin culture

Suction blisters were generated on neonatal C57BL/6 murine dorsal skin (P1) using a syringe and connector tubes (31). The back skin with the suction blister was excised and cultured with DMEM with 4.5 g/liter glucose and L-glutamine supplemented with 10% FBS and 1x antibiotic–antimycotic mixed stock solution. In the serum-starved condition, the culture medium was changed on the 2nd d of cultivation. In the serum-rich condition, the medium was replenished every day. Samples were fixed with formalin on the 4th d of cultivation.

## Live cell imaging

HaCaT cells at a concentration of $5.0 \times 10^5$ cells/ml were seeded onto $\mu$-Slide eight well chambers (250 $\mu$l per well) and 3.5 cm plastic dishes (2 ml per dish). HaCaT cells were cultured as indicated in the figures, and live cell imaging was carried out with a BZ-9000 or BZ-X800 microscope (Keyence) equipped with an incubation chamber maintained at 37°C and 5% $CO_2$.

## Autoperfusion system

HaCaT cells with a concentration of $5.0 \times 10^5$ cells/ml were seeded onto 3.5 cm plastic dishes (2 ml per dish). 2 d after seeding, the culture medium was replenished, and culture dishes were connected to the autoperfusion system. The autoperfusion system consisted of a micro tube pump system (iCOMES Lab) and 50 ml conical tubes that supplied the fresh culture medium and discarded the used culture medium. The perfusion rates were 2 ml/2 d for the low perfusion group and 6 ml/2 d for the high perfusion group, simulating serum-starved and serum-rich conditions, respectively. Samples were fixed with 4% PFA at RT for 10 min.

## Mathematical model

Consider a two-dimensional continuous model with cell density $\rho(x,y,t)$ and negative pressure $\sigma(x,y,t)$ (the diagonal component of the stress tensor) created by the accumulation of AJ complexes. The time evolution of these variables is governed by:

$$\frac{\partial \rho}{\partial t} = D_1 \nabla^2 \rho - \nabla \cdot \boldsymbol{J} + \eta$$

$$\frac{\partial \sigma}{\partial t} = D_2 \nabla^2 \sigma - \alpha\sigma + \beta\rho$$

where $\boldsymbol{J} = \lambda\rho\left(1 - \frac{\rho}{\rho^*}\right)\nabla\sigma$ represents the active cell flow because of force balance, which vanishes when the local cell density reaches the carrying capacity $\rho^*$. We set $\rho^* = 3$, allowing local squeezing and overlapping of cells of up to 300%. The variable $\eta$ accounts for the random motion of cells. To perform simulations, the two-dimensional space of size $[0, L] \times [0, L]$ was discretized into a 200 × 200 grid. To preserve the total cell volume, the random force at grid $(i,j)$ was chosen as $\eta(x_i,y_j,t) = N(\xi_{i+1,j}+\xi_{i-1,j}+\xi_{i,j+1}+\xi_{i,j-1}-4\xi_{i,j})$, where $N$ is the noise strength and $\xi_{i,j}$ is a Gaussian random variable with zero mean and unit variance. In this continuous model, the variation of the cell density $\rho$ corresponds to the cell shape change, where cells in low-$\rho$ regions are flattened and those in high-$\rho$ regions are compressed. Our model focuses solely on the patterning of cell density in the basal layer and does not consider stratification. The equation for stress ($\sigma$) is a natural extension of a one-dimensional model derived by reference 72, in which the off-diagonal component of the stress tensor is neglected. The stress $\sigma$ is assumed to be proportional to the concentration of AJs, which accumulates with the rate $\beta$ as the cell density increases, consistent with the observation that regions of high cell density form AJs in response to intercellular forces. It also decays at the rate $\alpha$. In a spatially homogeneous steady state ($\nabla^2\sigma = 0$ and $\partial\sigma/\partial t = 0$), the second equation reduces to $\sigma=(\beta/\alpha)\rho$: the stress proportional to the cell density. The time derivative of $\sigma$, which is absent in reference 72, was introduced so that the system relaxes to this steady state. The resulting model is a variant of the Keller–Segel system (73). A saturation term in the active flow in our model ensures that the density remains finite, as it diverges in the original Keller–Segel system. Although in the Keller–Segel system the variable $\sigma$ is interpreted as the concentration of a chemoattractant, whose production is proportional to the density $\rho$, our model was derived without assuming the existence of chemoattractants, where $\sigma$ is interpreted as stress.

For simulations, $\beta$ was chosen as a control parameter, and the other parameters were set as follows: $L = 200$, $D_1 = 0.5$, $D_2 = 5.0$, $\alpha = 1.0$, $\lambda = 1.0$, and $N = 0.5$ (for Fig 3) or $N \in \{0.1, 0.5, 1.0\}$ (for Fig S10). For all simulations, a spatially uniform initial condition $\rho(x,y,t) = 1$ and $\sigma(x,y,t) = 0$ was chosen (which means that cells are confluent in the system and no AJs are expressed) with the flux-free boundary condition.

### Animals

C57BL/6 mice were purchased from Clea Japan. The Institutional Review Board of the Hokkaido University Faculty of Medicine and Graduate School of Medicine approved all animal experiments in this study.

### Histology

For the epidermal thickness analysis, membrane samples from inserts or ex vivo cultured back skins were fixed with formalin and embedded in paraffin after dehydration. Thereafter, sectioned paraffin samples were deparaffinized and stained with hematoxylin and eosin (H&E). Images were taken with a BZ-9000 microscope, and the thickness of the epidermis was analyzed by ImageJ/Fiji (74).

### Immunofluorescent staining

The cells were cultured on $\mu$-Slide eight well chambers or $\mu$-Dish 35 mm dishes (ibidi). The cells were washed with PBS (Nacalai Tesque) and fixed with 4% PFA at RT for 10 min. Cells were permeabilized with 0.1% or 0.5% Triton X-100 in PBS for 20 min at RT, followed by blocking with 3% bovine serum albumin in PBS for 30 min. Subsequently, cells were incubated for 1–2 h at RT with the following primary antibodies: anti-E-cadherin antibody (Cat# 3195, RRID:AB_2291471, 1:100 dilution; Cell Signaling Technology), anti-$\alpha$18 antibody (17), anti-PH3 antibody (Cat# 07-554, RRID:AB_11210699, 1:250 dilution; Millipore), anti-$\alpha$-catenin antibody (Cat# 71-1200, RRID: AB_2533974, 1:100 dilution; Thermo Fisher Scientific), anti-K14 antibody (Cat# MA5-11599, RRID:AB_10982092, 1:100 dilution; Thermo Fisher Scientific), anti-K10 antibody (Cat# 905404, RRID:AB_2616955, 1:250 dilution; BioLegend), anti-YAP antibody (Cat# 14710, RRID: AB_2798583, 1:100 dilution; Cell Signaling Technology), and anti-ANKRD1 antibody (Cat# 67775-1g, AB_2918540, 1:500 dilution; Proteintech Group). Secondary antibodies, namely goat anti-mouse IgG Alexa Fluor 546 (Cat# A11003, RRID:AB_2534071, 1:1,000 dilution; Thermo Fisher Scientific) and goat anti-rabbit IgG Alexa Fluor 488 (A21206, RRID:AB_2535792, 1:1,000 dilution; Thermo Fisher Scientific), were incubated for 1 h at RT. Cells were washed in PBS 3 times for 5 min. Nuclei were stained with DAPI (Thermo Fisher Scientific), Hoechst 33342 (Dojindo), or propidium iodide (PI) (Dojindo) at a concentration of 0.5, 5, or 2 $\mu$g/ml, respectively, for 1 h at RT. Photo images were captured using BZ-9000 (Keyence), FV-1000 (Olympus), or LSM 710 (Zeiss) imaging systems. Phalloidin-iFluor 555 Reagent (Cat# ab176756, 1:500 dilution; Abcam) or wheat germ agglutinin (WGA) (Cat# W11263, 1:200 dilution; Thermo Fisher Scientific) were used for actin staining or membrane staining, respectively, at the time of primary antibody incubation. After washing with PBS, cells were observed with an LSM 710 microscope. In the autoperfusion experiment, cells were washed with PBS and stained with 0.5 $\mu$g/ml Hoechst 33342 at RT for 20 min. After PBS washes, images were taken using a BZ-9000 microscope. To observe the pattern of the 3D epidermis culture, the fixed membranes were washed with PBS and stained with 0.5 $\mu$g/ml Hoechst 33342 at RT for 1 h. Thereafter, inserts were washed with PBS and mounted in Fluoromount-G (Thermo Fisher Scientific). Images were obtained with an LSM 710 microscope.

### Western blot

WT or *CTNNA1*-KO HaCaT cells at a concentration of $1.0 \times 10^6$ cells/well were seeded onto six-well plastic plates. 2 d after seeding, we collected cell lysates using a lysis buffer containing 1% Nonidet P-40 (Nacalai Tesque), 25 mM Tris–HCl (pH 7.4), 100 mM NaCl, 10 mM ethylenediaminetetraacetic acid, and a 1:100 dilution protease inhibitor cocktail (P8340; Sigma-Aldrich) on ice for 30 min with shaking. The whole cell lysates were centrifuged at 15,300$g$ at 4°C for 20 min. Cell lysate supernatants were denatured with a 5× loading buffer (0.25 M Tris–HCl; 8% sodium dodecyl sulfate; 30% glycerol; 0.02% bromophenol blue; 0.3 M $\beta$-mercaptoethanol; pH 6.8). Samples were subjected to SDS–PAGE using NuPAGE 4 to 12%, Bis-Tris, 1.0–1.5 mm, Mini Protein Gels (Thermo Fisher Scientific). Proteins separated by SDS–PAGE were electrophoretically transferred onto PVDF transfer membranes (Bio-Rad). The membranes were blocked with 2% skim milk and incubated

with rabbit ant-α-catenin antibodies (Cat# 71-1200, RRID:AB_2533974, 1:200 dilution; Thermo Fisher Scientific) or rabbit anti-β-tubulin antibodies (Cat# ab6046, RRID:AB_2210370, 1:5,000 dilution; Abcam) in 2% skim milk at RT for 1 h. After washing with Tris-buffered saline, the membranes were incubated with a dilution of 1:5,000 peroxidase-conjugated anti-rabbit IgG antibody (AB_10015282; Jackson Immuno-Research) in 2% skim milk at RT for 1 h. Signals were visualized by Clarity Western ECL Substrate (Bio-Rad) and detected with an ImageQuant LAS 4000 mini camera system (Fujifilm).

## RNA-seq

HaCaT cells were seeded into 6 cm plastic dishes at a concentration of $2.5 \times 10^5$/ml or $7.5 \times 10^5$/ml. After overnight incubation at 37°C in humidified 5% $CO_2$, RNA was extracted using the FastGene RNA Premium Kit (Nippon Genetics) according to the manufacturer's instructions. Library preparation and sequencing were performed by Novogene. Briefly, mRNA was enriched using oligo(dT) beads. The mRNA was fragmented randomly by adding a fragmentation buffer, then the cDNA was synthesized using an mRNA template and a random hexamer primer, after which a custom second-strand synthesis buffer (Illumina), dNTPs, Rnase H, and DNA polymerase I were added to initiate second-strand synthesis. After a series of terminal repair, A-tailing, and sequencing adaptor ligation, the double-stranded cDNA library was completed through size selection and PCR enrichment. After the quality control of the libraries, the libraries were sequenced by the NovaSeq 6000 system in PE150 mode. Reads were then mapped to hg38 using STAR (v.2.7.3) (75). Gene expression levels were quantified using RSEM (v.1.3.1) (76). Read counts were analyzed through integrated differential expression and pathway analysis (iDEP9.51) (77). Genes with low-level expression (less than 0.5 counts per million in all samples) were removed from the analysis. Genes that were differentially expressed between low-density and high-density conditions were identified with DESeq2 using a threshold of false discovery rate of <0.1 and a fold-change of >2 (16). The up-regulated and down-regulated genes were subjected to enrichment analysis.

## Fluorescent intensity analysis

Intensity ratios of YAP and ANRKD1 were calculated using the intensities of the nucleus regions and regions adjacent to the nucleus using Fiji (74). Each image included three to six measurement spots, and each plot represents the mean calculated from two to four images. The intensities of E-cadherin and α-catenin at cell–cell adhesion sites were quantified by the line plot function of Fiji (74). Initially, cell–cell adhesion sites were identified using WGA staining. Subsequently, vertical line profiles were drawn at these sites, and the distance of 0 μm was set at the peak of each fluorescence intensity plot. The intensities of α18 were plotted using the same line profile of α-catenin with distances calibrated according to α-catenin. Ten line profiles per image were analyzed, and the mean intensity at each distance was calculated. Representative intensity profiles were depicted as mean ± SEM, based on data from four images. For comparative analysis of intensity ratios between high- and low-density areas, mean ratios were calculated as relative values to the intensities in low-density areas, derived from six independent experiments.

## Calculation of pattern index

Grayscale versions for immunofluorescent images of keratinocytes, in which the nuclei were labeled with Hoechst 33342, were used to evaluate the inhomogeneity of cell density. The image was binarized with the threshold value determined from its brightness value histogram (i.e., the threshold value was defined as the brightness that had the maximum count and was greater than 0.6 times the average brightness of the total image). Subsequently, the binarized image was blurred using a linear filter within a circle with a radius R. Here, R was set to four pixels, corresponding to ~10 μm, which was nearly the same as the distance between neighboring cells. It should be noted that the blurred image was almost uniform if the cells were uniformly distributed, but it had different values if the cell density changed. Thus, the standard of the blurred images was calculated and adopted as the pattern index, in which a greater value indicates a more spatially inhomogeneous distribution of cells.

## Evaluation of characteristic length of the pattern

Grayscale versions for immunofluorescent images of keratinocytes, in which the nuclei were labeled with Hoechst 33342, were used to evaluate the characteristic length of the spatial pattern. First, the images were processed with a band-pass filter with the range of 5 to 200 pixels (~26.5–1,060 μm) so that the large-scale gradient of brightness because of the inhomogeneous illumination and small-scale structure below the size of cells would not affect the evaluation results. Then, the radial autocorrelation function $G(r)$ was calculated as

$$G(r) = \frac{1}{2\pi} \int_0^{2\pi} g(r\cos\theta, r\sin\theta) \, d\theta$$

where

$$g(x,y) = \frac{1}{A} \iint \{f(x', y') - \overline{f}\}\{f(x + x', y + y') - \overline{f}\} \, dx \, dy$$

Here, $f(x,y)$ is the brightness of the processed image at the coordinates $(x,y)$, $\overline{f}$ is its averaged brightness, and $A$ is its area. The value of $r$ at the first peak of $G(r)$ in $r > 0$ corresponds to the characteristic length of the pattern; that is, the distance between the neighboring regions with high cell densities.

## Statistical analysis

Statistical analyses were performed with GraphPad Prism 9 (GraphPad Software). P-values were determined with two-tailed Mann–Whitney U tests, Kruskal–Wallis tests followed by Dunn's multiple comparison test, or Wilcoxon matched-pairs signed rank test. P-values of < 0.05 were considered statistically significant.

# Data Availability

The datasets produced in this study are available in the following databases: RNA-Seq data: Gene Expression Omnibus GSE247733.

# Supplementary Information

# Acknowledgements

We thank Nami Ikeshita and Mika Tanabe for their technical assistance. We also thank Professor Akira Nagafuchi for providing the α18 antibody. This work was funded by JSPS KAKENHI Grant Number 23K15277, the Tokyo Biochemical Research Foundation, the Nakatomi Foundation to Y Mai, JSPS KAKENHI Grant Number JP22K03428 to Y Kobayashi, JST CREST Grant Number JPMJCR1926, JSPS KAKENHI Grant Number JP23H04936 and JP23K20808 to M Nagayama, the Project for Junior Scientist Promotion of Hokkaido University to H Ujiie, JSPS KAKENHI Grant Number 23H02928, the Akiyama Life Science Foundation, and the Mochida Memorial Foundation for Medical and Pharmaceutical Research to K Natsuga.

## Author Contributions

Y Mai: conceptualization, data curation, formal analysis, funding acquisition, validation, investigation, visualization, methodology, and writing—original draft.

Y Kobayashi: conceptualization, data curation, formal analysis, funding acquisition, investigation, visualization, methodology, and writing—original draft.

H Kitahata: software, formal analysis, visualization, methodology, and writing—original draft.

T Seo: investigation, methodology, and writing—review and editing.

T Nohara: investigation, methodology, and writing—review and editing.

S Itamoto: investigation and writing—review and editing.

S Mai: investigation and writing—review and editing.

J Kumamoto: methodology and writing—review and editing.

M Nagayama: conceptualization, funding acquisition, methodology, and writing—review and editing.

W Nishie: conceptualization, data curation, formal analysis, investigation, methodology, and writing—review and editing.

H Ujiie: conceptualization, resources, supervision, funding acquisition, methodology, project administration, and writing—review and editing.

K Natsuga: conceptualization, data curation, supervision, funding acquisition, methodology, project administration, and writing—review and editing.

## Conflict of Interest Statement

The authors declare that they have no conflict of interest.

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
