## [Reviewer comments · Life Science Alliance]

Life Science Alliance

Patterning in stratified epithelia depends on cell-cell adhesion

Yosuke Mai, Yasuaki Kobayashi, Hiroyuki Kitahata, Takashi Seo, Takuma Nohara, Sota Itamoto, Shoko Mai, Junichi Kumamoto, Masaharu Nagayama, Wataru Nishie, Hideyuki Ujije, and Ken Natsuga

DOI: <https://doi.org/10.26508/lsa.202402893>

Corresponding author(s): Ken Natsuga, Hokkaido University

Review Timeline:

Submission Date:	2024-06-19
Editorial Decision:	2024-06-24
Revision Received:	2024-06-25
Accepted:	2024-06-26

Transaction Report:

Please note that the manuscript was reviewed at Review Commons and these reports were taken into account in the decision-making process at *Life Science Alliance*.

Review
COMMONS

Reviews

Review #1

****Summary****

In this study, the researchers investigate the spontaneous patterning of keratinocytes. As model they use HaCaT cells, an immortalized keratinocyte line. The cells exhibit a self-organized pattern of high and low cell density, which is disrupted by medium changes but reappear over time. The researchers find that serum starvation and high calcium concentration are crucial for the formation of these keratinocyte patterns.

RNA sequencing analysis of regions of high vs low density indicates enrichment in gene ontology terms related to cell-cell adhesion, mainly adherens junctions (AJs), and keratinocyte differentiation. Experimental manipulations, such as inhibiting E-cadherin- or α -catenin-mediated adhesion, and disrupting myosin-II activity, all interfere with the formation of keratinocyte patterns, emphasizing the importance of AJs.

Mathematical modeling suggests that cell-cell adhesion alone is sufficient for the emergence of density patterns.

Keratinocyte patterns have spatial regulation of keratinocyte differentiation and proliferation. Differentiated cells are abundant in areas of high cell density, while proliferative cells are in areas of low cell density. The authors verify that YAP activity regulates pattern-dependent differentiation and proliferation.

The role of serum starvation and cell-cell adhesion through AJs in the differentiation of keratinocytes are supported by epidermal stratification experiments in 3D culture, and ex vivo experiments on mouse skin suction blister wounds.

In conclusion, the study provide insights into the spatial regulation of differentiation and proliferation in epidermal cells.

****Major comments****

Although not novel, given that it has been already demonstrated with several other epithelial cell monolayers and in vivo in *Drosophila*, the conclusions that serum starvation facilitates epidermal stratification through cell-cell adhesion is convincing. It is unclear whether the cell patterning the authors are describing is a real patterning, defined in biology as any regularly repeated cell or structural arrangement or simply an inhomogeneous distribution of cell densities. The conclusion that the cell-cell adhesion signaling pathway identified in the paper "might promote wound healing in clinical settings" (last sentence of the abstract) is not substantiated by the results.

It would be opportune to better describe the type of "cell patterning" that the authors are seeing in their experiments. In my opinion the effect seen in the described experiment is not a "patterning" but a difference in cell density which can be less or more homogeneous in an HaCat monolayer.

Importantly, it is unclear whether the "cell patterning" is a subsequent consequence or proceed stratification. It is unclear how starvation relates to the increased adhesions and YAP signaling. The authors conclude the discussion section proposing "that molecules involved in cell-cell adhesion-induced patterning are suitable target candidates to facilitate wound healing". None the experiments done in the wound healing setting are addressing the role of any molecules described in the paper. I would suggest the authors to remove this last claim from the manuscript. Alternatively, the authors should provide evidence that targeting some of the molecules described in the manuscript are accelerating wound healing in a clinically relevant model of wound healing.

I would request the authors to provide the following essential data to substantiate their experiments:

- Provide a full gene list related to Figure 2a.
- In relation to Figure 2c, stain for α -catenin and quantify the intensity ration of α -catenin vs α -18-catenin as proper readout of adhesion strength (see Yonemura et al., Nat Cell Biol 2010).
- Properly quantify nuclear vs cytoplasmic localization of YAP in low vs high density areas in Figure 4f.
- The nuclear localization of YAP is not sufficient to demonstrate activation of the YAP signaling. The authors should provide evidence of YAP activity in low vs high density areas looking for example at known downstream target genes in epithelial cells (see Zhao et al., Genes Dev 2007; Yu et al., Cell 2012; Aragona et al., Cell 2013).
- The activity of PY-60 in Figure 4g and XAV939 in Figure 4i as YAP activator and repressor respectively, should be controlled against YAP localization and activity.
- In Figure 5a a quantification of the numbers of cell layers should be used instead of the thickness and a staining and quantification of K14 and K10 should be added to formally address stratification.

Most of the proposed experiments are simply additional quantifications of images or adjustments of data that are already available to the authors. I estimate that the remaining experiments can be done in less than a month and will not require additional expertise.

The methods, figures presentation and legends, and the statistical analysis are adequate, clear and accurate.

****Minor comments****

There are three fundamental studies that the authors should discuss:

- Saw, Doostmohammadi et al., Nature 2017. Topological defects in epithelia govern cell death and extrusion. Here, the role of topological defects (see also Bonn et al., Phys Res E 2022) and a-catenin-dependent cell-cell interactions are connected to cell extrusion and Yap activity in epithelial monolayers including HaCat cells.
- Miroshnikova et al., Nat Cell Biol 2018. Adhesion forces and cortical tension couple cell proliferation and differentiation to direct epidermal stratification. Here, the authors demonstrated that the increase of cell-cell adhesion couples with a decrease of cortical tension triggers stratification in the skin epidermis.
- Boockook et al., Nature Physics 2021. Theory of mechanochemical patterning and optimal migration in cell monolayers. Here, cell density and ERK activity are formalized to be key players in patterning formation in a cell monolayer.

In addition, several components of the Hippo-YAP pathway are known regulators of cell-cell adhesion (e.g. AMOT and NF2) and should be discussed (for reference see reviews on the topic Zheng & Pan, Dev Cell 2019; Karaman & Halder Cold Spring Harb Perspect Biol 2018; Gumbiner & Kim, J Cell Sci 2014) as important molecules implicated in the biological phenomena described in the manuscript.

The study aims at understanding spontaneous patterning of keratinocytes. The authors nicely employ various experimental approaches, including cell imaging, RNA sequencing, cell manipulation by genetic engineering and pharmacological treatments, and mathematical modeling, to elucidate the underlying cellular and molecular mechanisms regulating this process. However, several of the conclusions presented in the manuscript do not present any conceptual advance to the field of self-organization of cell density patterns or epithelial biology.

The role of starvation in effecting epithelial growth is very well known. The role of AJ in pattern formation has been described previously in epithelial monolayers (Saw, Doostmohammadi et al., Nature 2017) and in vivo in Drosophila (Mao et al., Genes Dev 2011; Mao et al., EMBO J 2013). The effect of cell density on YAP signaling is known (Zhao et al., Genes Dev 2007; Aragona et al., Cell 2013). The importance of AJ for keratinocytes differentiation and stratification has been demonstrated in vitro and in vivo (Miroshnikova et al., Nat Cell Biol 2018). The role of a-catenin upstream of YAP activity in regulating interfollicular epidermis stem cells self-renewal and wound healing has been demonstrated in vitro and in vivo by the group of Fernando Camargo in Cell 2011.

The manuscript could be of interest for researchers interested in basic cell biology and a specialised audience in cell self-organisation.

My field of expertise: epithelial biology, stem cell biology, skin homeostasis and wound healing, mechanobiology, YAP signaling. I do not have sufficient expertise to evaluate the mathematical modelling.

Review #2

****Summary:****

Mai et al. reported an interesting observation that serum starvation induced the keratinocytes, a type of epithelial cells, to form a pattern characterized by regions with high and low densities. They showed that this patterning processing depends on cell-cell adhesion using a series of pharmacological treatment and a CRISPR knockout of alpha-catenin. They used mathematical modeling to demonstrate that cell-density dependent stress can sufficiently generate patterns of high and low cell densities, but the interpretation of the modeling is questionable (see below). They showed correlation of a differentiated keratinocyte marker, keratin 10, with the high-density region, but over claimed this result as patterning modulates differentiation. They also showed correlation of YAP activity (cytoplasmic to nuclear ratio) to the high vs. low-density regions. Interestingly, treatment with a YAP activator PY-60 disrupted pattern formation, while the YAP inhibitor XAV939 barely affected pattern formation. Finally, the authors demonstrated that serum starvation increased the thickness of keratinocytes cultured in a trans-well system (which they called 3D culture), and a mouse back skin explant compared to serum-rich culture conditions. In the former system, they showed dependence on alpha-catenin using the CRISPR knockout.

****Major comments:****

The conclusion that "mathematical modeling indicates that cell-cell adhesion alone is sufficient to form regions with high/low cell density" is misleading. The key assumption of the modeling is that the time derivative of stress (d_{σ}/dt) is proportional to the cell density (ρ), where the proportion parameter (β) was interpreted as cell adhesion strength. However, β could be interpreted as any general attractor proportional to the cell density, such as a chemoattractant. In addition, it is unclear why the time derivative of stress (d_{σ}/dt) instead of stress itself (σ) is proportional to the cell density. The authors should further clarify the meanings of modeling parameters and be more careful with their conclusions.

Related to above, the authors should revise the title to reflect that the patterning depends on cell-cell adhesion instead of claiming that cell-cell adhesion drives patterning. This would require experimentally demonstrating sufficiency, for

example, showing that increasing adhesion in a cell line with low adhesion that does not show patterning can sufficiently induce patterning.

The conclusion that "patterning modulates differentiation" is not supported by evidence. Differentiation as evidenced by the presence of keratin 10 occurred as early as day 2 before any signs of patterning (Fig. 4A). When patterning was completely disrupted by alpha-catenin KO, there are still many keratin 10 positive cells. The apparent higher proportion of keratin 10+ cells in the wild type seems to be merely reflecting the higher cell density - if the quantification were normalized by the cell number, they are probably comparable. Overall, the presented data only supports a correlation of the differentiation marker keratin 10 with high-density regions.

The choice of RNA-seq comparison groups (high-density vs. low-density culture) is puzzling, since the effects caused by culture density changes may not be related to the high vs. low-density regions in the patterned cultures. There are so many changes there and the rationale of following up on cell adhesion was unclear. In fact, it seems that the RNA-seq data didn't help the logic flow of the paper at all.

The claim of 3D culture of keratinocytes is confusing. The culture in the trans-well insert is still on the flat 2D surface, why should it be called 3D culture? If the point is to culture at air/liquid interface, that should instead be emphasized instead of calling it 3D.

The observation that serum starvation and replenishment induced reversible patterning of the keratinocytes is quite interesting. However, the biological relevance is unclear - isn't all skin stratified? The evidence supporting the dependence of this patterning on adherens junction by disrupting E-cadherin, myosin, or alpha-catenin is convincing, although not surprising. The involvement of YAP in differentiation vs. proliferation is interesting, but it's in line with the known functions of YAP. The modeling part, with some clarification, can be quite insightful. Overall, this research could be interesting to those working in epithelial morphogenesis, if further developed.

My expertise is in epithelial tissue morphogenesis, mechanobiology, and extracellular matrix biology.

Review #3

In this manuscript the authors aim to understand the signals that coordinate spatial patterns of keratinocyte proliferation and differentiation. To address this question the authors use the HaCaT keratinocyte cell line that upon serum starvation forms spatially separated domains of proliferation and differentiation. The data presented in this manuscript potentially suggest that serum starvation works through adherens junctions to create differentially dense fields within the cultures which determines whether cells proliferate or differentiate. The authors then perform experiments to show that junction formation with starvation drive keratinocyte differentiation potentially through YAP signaling. However, these experiments are rather loosely connected and their results often do not support the conclusions drawn by the authors. However, the not well supported conclusion forms the basis for a fact statement, but their data really did not show that. For example, the authors state: "By contrast, YAP inhibition by a tankyrase inhibitor, XAV939, suppressed pattern-dependent proliferation (Fig. 4i, j)," . However, their data do not show that proliferation is pattern-dependent but is nevertheless used to connect to and draw a conclusion about YAP signaling. The data itself appear to be of high quality, figures are well organized and statistics of quantification seem appropriate, but it is somewhat problematic that throughout the manuscript it remains unclear if certain statements are hypotheses or conclusions on real data. Pattern formation as a requirement for differentiation is an interesting concept. However, the presented study lacks proper conclusive data how these patterns may contribute to proliferation and differentiation and remains rather short on what exactly is the instructive nature of these patterns, as they only use high density and are not generating own patterns with defined cues that explore what cues contribute.

****Major points:****

The statement "According to the RNA-seq data, AJ molecules, such as E-cadherin and actin, were localized at intercellular junctions in areas of high cell density" is not correct. RNA-seq does not allow conclusions about protein localization. Instead, the GO-Term analysis shown in Figure 2b shows downregulation of "cell-adhesion" in dense areas. Consistently the E-cadherin staining presented in Figure 2c suggests lower intercellular E-cadherin levels in the most dense areas. However, any statement about junctional localization of adhesion components requires e.g. intensity quantification at junctions vs. cytoplasm or else to discriminate from intense overall staining due to high cell density and thus high overall junction numbers. Hence, even though potentially true, the statement: "These data suggest that cells in regions of high cell density form AJs in response to intercellular forces" is not fully supported by the data shown so far.

The authors suggest a pattern of high and low density that is formed over time. However, at the same time high density areas show formation of a second layer. Hence, "denser" areas as observed by phase contrast images or DAPI positive nuclei may either represent dense or stratified cells. What is missing is an analysis of cell density before cells started to stratify making sure only cells in the basal layer are analyzed. Otherwise, density and stratification which are perhaps interdependent in this system cannot be discriminated.

The mathematical model does not include stratification and it is thus not clear to what extent it may explain the observed patterns. Moreover, the model appears to assume variables that have not been determined or cited. This reviewer is not an expert in modeling and thus cannot fully judge the math behind the model. However, the model appears to be biased if it assumes, as mentioned, that cell-adhesion increases with density. If low adhesion forces do not produce patterns, what is the counterforce in the model? Are cells allowed to change size to enable low density areas or do cells lose contact with neighbors despite high adhesion strength? Overall, it appears that the model is set up such, that it tends to reproduce what was observed in experiment. This conclusion, however, may result of an incomplete understanding of the model parameters.

If dense areas do actually represent stratified areas it may not be surprising that the GO analysis indicates an increase in differentiation. A requirement for AJ or intercellular junctions in general is less surprising as stratification requires cell-cell adhesion. The observation that AJ are essential for intercellular junction formation in keratinocytes or in other epithelial cells is not new (e.g. Michels et al. JID 2009).

The part of the paper addressing the role of YAP suffers from a number of potentially mislead assumptions/conclusions based on a previous experiment which then did not properly supported that conclusion (see also overall comments). For example, the statement "YAP inhibition by a tankyrase inhibitor, XAV939, suppressed pattern-dependent proliferation" contains interdependencies that have not been show. XAV939 may just inhibit proliferation which is not necessarily pattern dependent. Too much speculation confuses data and hypotheses.

The 3D HaCaT cultures are performed on transwell filters with medium supply above and below cells, with the assumption that organizing patterns are also formed under these conditions. However, this has not been shown by the authors. Their suggestion that serum starvation may increases thickness of cultures through alterations in the organization of

The mechanisms that drive self-organization of epithelial cells to spatially separate domains of proliferation and differentiation is in principle a very interesting topic of high interest to the cell and mechanobiology community,

Authors' Response to Reviewers

1. Point-by-point description of the revisions

This section is mandatory. Please insert a point-by-point reply describing the revisions that were already carried out and included in the transferred manuscript.

Reviewer #1 (Evidence, reproducibility and clarity (Required)):

SUMMARY

In this study, the researchers investigate the spontaneous patterning of keratinocytes. As model they use HaCaT cells, an immortalized keratinocyte line. The cells exhibit a self-organized pattern of high and low cell density, which is disrupted by medium changes but reappear over time. The researchers find that serum starvation and high calcium concentration are crucial for the formation of these keratinocyte patterns.

RNA sequencing analysis of regions of high vs low density indicates enrichment in gene ontology terms related to cell-cell adhesion, mainly adherens junctions (AJs), and keratinocyte differentiation.

Experimental manipulations, such as inhibiting E-cadherin- or α -catenin-mediated adhesion, and disrupting myosin-II activity, all interfere with the formation of keratinocyte patterns, emphasizing the importance of AJs.

Mathematical modeling suggests that cell-cell adhesion alone is sufficient for the emergence of density patterns.

Keratinocyte patterns have spatial regulation of keratinocyte differentiation and proliferation. Differentiated cells are abundant in areas of high cell density, while proliferative cells are in areas of low cell density.

The authors verify that YAP activity regulates pattern-dependent differentiation and proliferation.

The role of serum starvation and cell-cell adhesion through AJs in the differentiation of keratinocytes are supported by epidermal stratification experiments in 3D culture, and ex vivo experiments on mouse skin suction blister wounds.

In conclusion, the study provide insights into the spatial regulation of differentiation and proliferation in epidermal cells.

MAJOR COMMENTS

*Although not novel, given that it has been already demonstrated with several other epithelial cell monolayers and in vivo in *Drosophila*, the conclusions that serum starvation facilitates epidermal stratification through cell-cell adhesion is convincing. It is unclear whether the cell patterning the authors are describing is a real patterning, defined in biology as any regularly repeated cell or structural arrangement or simply an inhomogeneous distribution of cell densities.*

We have addressed this issue by analyzing our images with the autocorrelation function (see Fig. 1g, 1h, and Supplementary Fig. 5) and confirmed that the distribution of high/low cell density is patterned with the average nearest neighbor distance between areas of high cell density being approximately 300 μm . We have incorporated these new data into the revised manuscript.

The conclusion that the cell-cell adhesion signaling pathway identified in the paper "might promote wound healing in clinical settings" (last sentence of the abstract) is not substantiated by the results.

We agree with the reviewer's point and have deleted the sentence in the abstract, accordingly.

It would be opportune to better describe the type of "cell patterning" that the authors are seeing in their experiments. In my opinion the effect seen in the described experiment is not a "patterning" but a difference in cell density which can be less or more homogeneous in an HaCat monolayer.

Please see the answer above on our analysis using the autocorrelation function.

Importantly, it is unclear whether the "cell patterning" is a subsequent consequence or proceed stratification.

As the mathematical modeling indicated patterning without the need for stratification steps, we believe that cell patterning is not a direct consequence of stratification. However, it is technically difficult to differentiate whether patterning developed prior to stratification in our experimental settings. We have added this limitation to the Discussion of the revised manuscript.

It is unclear how starvation relates to the increased adhesions and YAP signaling.

As the reviewer pointed out, we could not address what molecules in the serum are responsible because the serum is a complex mixture of biomolecules that includes hormones, growth factors, vitamins, and other nutrients. We have added this limitation of our study to the revised manuscript.

The authors conclude the discussion section proposing "that molecules involved in cell-cell adhesion-induced patterning are suitable target candidates to facilitate wound healing". None the experiments done in the wound healing setting are addressing the role of any molecules described in the paper. I would suggest the authors to remove this last claim from the manuscript. Alternatively, the authors should provide evidence that targeting some of the molecules described in the manuscript are accelerating wound healing in a clinically relevant model of wound healing.

We agree with the reviewer's point and have deleted the passage in the revised manuscript, accordingly.

*I would request the authors to provide the following essential data to substantiate their experiments:
- Provide a full gene list related to Figure 2a.*

We have provided the gene list (Supplementary Table 1), accordingly.

- In relation to Figure 2c, stain for α -catenin and quantify the intensity ration of α -catenin vs α -18-catenin as proper readout of adhesion strength (see Yonemura et al., Nat Cell Biol 2010).

As the reviewer pointed out, the intensity ratio of α -catenin vs. α 18 is a general readout of cell adhesion strength. However, this ratio should be based on similar intensity of alpha catenin between two groups for comparison. In contrast, the intensity of α -catenin itself was weaker in the area with low cell density compared with in that with high cell density in our experimental setting (Supplementary Fig. 8d, e, g), which could greatly affect the ratio. To overcome this problem, we have reanalyzed line plots of α -catenin immunofluorescence, picked up the α 18 intensity at the peaks (corresponding to cell-cell adhesion) of α -

catenin, and compared that of high and low cell density area. As expected, α 18 was more pronounced in the area with high cell density. We have added the data to Supplementary Fig. 8d-h in the revised manuscript.

- Properly quantify nuclear vs cytoplasmic localization of YAP in low vs high density areas in Figure 4f.

According to the reviewer's suggestion, we have quantified nuclear/cytoplasmic YAP and added the data (Revised Fig. 6b (original Fig. 4)) to the revised manuscript.

- The nuclear localization of YAP is not sufficient to demonstrate activation of the YAP signaling. The authors should provide evidence of YAP activity in low vs high density areas looking for example at known downstream target genes in epithelial cells (see Zhao et al., Genes Dev 2007; Yu et al., Cell 2012; Aragona et al., Cell 2013).

We have analyzed ANKRD1 (Yu et al., Cell 2012) as a YAP readout molecule and confirmed that, in line with YAP dynamics, ANKRD1 was localized in the nucleus of high cell density area. We have provided the data (Revised Fig. 6c, d (original Fig. 4)) for the revised manuscript.

- The activity of PY-60 in Figure 4g and XAV939 in Figure 4i as YAP activator and repressor respectively, should be controlled against YAP localization and activity.

We have quantitatively analyzed YAP and ANKRD1 localization upon chemical treatment and added the data (Supplementary Fig. 1a-d, g-j (original Supplementary Fig. 8)) to the revised manuscript.

- In Figure 5a a quantification of the numbers of cell layers should be used instead of the thickness and a staining and quantification of K14 and K10 should be added to formally address stratification.

As expected, the number of K10-positive cell layers was larger in serum-starved conditions than in serum-rich conditions, while the number of K14-positive cell layer was comparable between the two groups. We have provided the quantification data (Supplementary Fig. 12 c-e (original Supplementary Fig. 9)) to the revised manuscript accordingly.

Most of the proposed experiments are simply additional quantifications of images or adjustments of data that are already available to the authors. I estimate that the remaining experiments can be done in less than a month and will not require additional expertise.

The methods, figures presentation and legends, and the statistical analysis are adequate, clear and accurate.

MINOR COMMENTS

There are three fundamental studies that the authors should discuss:

- Saw, Doostmohammadi et al., Nature 2017. Topological defects in epithelia govern cell death and extrusion. Here, the role of topological defects (see also Bonn et al., Phys Res E 2022) and a-catenin-

dependent cell-cell interactions are connected to cell extrusion and Yap activity in epithelial monolayers including HaCat cells.

- Miroshnikova et al., Nat Cell Biol 2018. Adhesion forces and cortical tension couple cell proliferation and differentiation to direct epidermal stratification. Here, the authors demonstrated that the increase of cell-cell adhesion couples with a decrease of cortical tension triggers stratification in the skin epidermis.

- Boockock et al., Nature Physics 2021. Theory of mechanochemical patterning and optimal migration in cell monolayers. Here, cell density and ERK activity are formalized to be key players in patterning formation in a cell monolayer.

In addition, several components of the Hippo-YAP pathway are known regulators of cell-cell adhesion (e.g. AMOT and NF2) and should be discussed (for reference see reviews on the topic Zheng & Pan, Dev Cell 2019; Karaman & Halder Cold Spring Harb Perspect Biol 2018; Gumbiner & Kim, J Cell Sci 2014) as important molecules implicated in the biological phenomena described in the manuscript.

We appreciate the reviewer's suggestion and have cited and discussed these seminal papers in the revised manuscript.

Reviewer #1 (Significance (Required)):

The study aims at understanding spontaneous patterning of keratinocytes. The authors nicely employ various experimental approaches, including cell imaging, RNA sequencing, cell manipulation by genetic engineering and pharmacological treatments, and mathematical modeling, to elucidate the underlying cellular and molecular mechanisms regulating this process. However, several of the conclusions presented in the manuscript do not present any conceptual advance to the field of self-organization of cell density patterns or epithelial biology.

The role of starvation in effecting epithelial growth is very well known. The role of AJ in pattern formation has been described previously in epithelial monolayers (Saw, Doostmohammadi et al., Nature 2017) and in vivo in Drosophila (Mao et al., Genes Dev 2011; Mao et al., EMBO J 2013). The effect of cell density on YAP signaling is known (Zhao et al., Genes Dev 2007; Aragona et al., Cell 2013). The importance of AJ for keratinocytes differentiation and stratification has been demonstrated in vitro and in vivo (Miroshnikova et al., Nat Cell Biol 2018). The role of a-catenin upstream of YAP activity in regulating interfollicular epidermis stem cells self-renewal and wound healing has been demonstrated in vitro and in vivo by the group of Fernando Camargo in Cell 2011.

The manuscript could be of interest for researchers interested in basic cell biology and a specialised audience in cell self-organisation.

My field of expertise: epithelial biology, stem cell biology, skin homeostasis and wound healing, mechanobiology, YAP signaling. I do not have sufficient expertise to evaluate the mathematical modelling.

We appreciate the reviewer's constructive comments.

Reviewer #2 (Evidence, reproducibility and clarity (Required)):

Summary:

Mai et al. reported an interesting observation that serum starvation induced the keratinocytes, a type of epithelial cells, to form a pattern characterized by regions with high and low densities. They showed that this patterning processing depends on cell-cell adhesion using a series of pharmacological treatment and a CRISPR knockout of alpha-catenin. They used mathematical modeling to demonstrate that cell-density dependent stress can sufficiently generate patterns of high and low cell densities, but the interpretation of the modeling is questionable (see below). They showed correlation of a differentiated keratinocyte marker, keratin 10, with the high-density region, but over claimed this result as patterning modulates differentiation. They also showed correlation of YAP activity (cytoplasmic to nuclear ratio) to the high vs. low-density regions. Interestingly, treatment with a YAP activator PY-60 disrupted pattern formation, while the YAP inhibitor XAV939 barely affected pattern formation. Finally, the authors demonstrated that serum starvation increased the thickness of keratinocytes cultured in a trans-well system (which they called 3D culture), and a mouse back skin explant compared to serum-rich culture conditions. In the former system, they showed dependence on alpha-catenin using the CRISPR knockout.

Major comments:

The conclusion that "mathematical modeling indicates that cell-cell adhesion alone is sufficient to form regions with high/low cell density" is misleading. The key assumption of the modeling is that the time derivative of stress (d_sigma/dt) is proportional to the cell density (ρ), where the proportion parameter (β) was interpreted as cell adhesion strength. However, β could be interpreted as any general attractor proportional to the cell density, such as a chemoattractant.

Our purpose here is to demonstrate that the model based on the assumption of cell-cell adhesion as a mere source of attractive forces can reproduce the experimentally observed spatial patterning. As the referee rightly points out, the term $\beta\rho$ in the second equation allows different interpretations such as the effect of attractant proportional to cell density. Therefore, our mathematical model cannot be used as a proof of the existence of cell-cell adhesion. We have reduced the tone in the revised manuscript.

In addition, it is unclear why the time derivative of stress (d_sigma/dt) instead of stress itself (σ) proportional to the cell density. The authors should further clarify the meanings of modeling parameters and be more careful with their conclusions.

If the system is in the steady state ($d_sigma/dt = 0$) with no spatial variations ($\nabla^2 \sigma = 0$), then the second equation reduces to $\sigma = (\beta/\alpha) \rho$, namely that the cell density is proportional to stress, as pointed out by the referee.

Our model, which describes temporal and spatial variations, generalizes this situation. The spatial dependence represented by $\nabla^2 \sigma$ was introduced according to the Reference 72 (original Reference 51). Furthermore, we introduced the time derivative d_sigma/dt to account for the fact that the system should relax into the steady state described above. We have included these into the revised manuscript.

Related to above, the authors should revise the title to reflect that the patterning depends on cell-cell adhesion instead of claiming that cell-cell adhesion drives patterning. This would require experimentally

demonstrating sufficiency, for example, showing that increasing adhesion in a cell line with low adhesion that does not show patterning can sufficiently induce patterning.

We agreed with the reviewer and have revised the title into "Patterning in stratified epithelia depends on cell-cell adhesion" and reduce the tone of the final sentence of the Discussion section accordingly.

The conclusion that "patterning modulates differentiation" is not supported by evidence. Differentiation as evidenced by the presence of keratin 10 occurred as early as day 2 before any signs of patterning (Fig. 4A). When patterning was completely disrupted by alpha-catenin KO, there are still many keratin 10 positive cells. The apparent higher proportion of keratin 10+ cells in the wild type seems to be merely reflecting the higher cell density - if the quantification were normalized by the cell number, they are probably comparable. Overall, the presented data only supports a correlation of the differentiation marker keratin 10 with high-density regions.

According to the reviewer's suggestion, we have reduced the tone of the title of Revised Fig. 4 (original Fig. 3) and changed it into "Patterning correlates with differentiation and proliferation markers in keratinocytes".

The choice of RNA-seq comparison groups (high-density vs. low-density culture) is puzzling, since the effects caused by culture density changes may not be related to the high vs. low-density regions in the patterned cultures. There are so many changes there and the rationale of following up on cell adhesion was unclear. In fact, it seems that the RNA-seq data didn't help the logic flow of the paper at all.

Although we believe that comparison between high-density and low-density culture partly recapitulates high/low cell density regions in our study, the comparison is not identical to patterned cultures as the reviewer pointed out. We have moved RNA-seq data to the Supplementary Information (Supplementary Fig. 7) and added more analysis to address that cell adhesion and differentiation are major differences between high-density and low-density culture, supporting further analysis on this matter in our study.

The claim of 3D culture of keratinocytes is confusing. The culture in the trans-well insert is still on the flat 2D surface, why should it be called 3D culture? If the point is to culture at air/liquid interface, that should instead be emphasized instead of calling it 3D.

We have changed "3D culture" into air-liquid interface culture, accordingly.

Reviewer #2 (Significance (Required)):

The observation that serum starvation and replenishment induced reversible patterning of the keratinocytes is quite interesting. However, the biological relevance is unclear - isn't all skin stratified? The evidence supporting the dependence of this patterning on adherens junction by disrupting E-cadherin, myosin, or alpha-catenin is convincing, although not surprising. The involvement of YAP in differentiation vs. proliferation is interesting, but it's in line with the known functions of YAP. The modeling part, with some clarification, can be quite insightful. Overall, this research could be interesting to those working in epithelial morphogenesis, if further developed.

My expertise is in epithelial tissue morphogenesis, mechanobiology, and extracellular matrix biology.

We appreciate the reviewer's constructive and thoughtful comments.

Reviewer #3 (Evidence, reproducibility and clarity (Required)):

In this manuscript the authors aim to understand the signals that coordinate spatial patterns of keratinocyte proliferation and differentiation. To address this question the authors use the HaCaT keratinocyte cell line that upon serum starvation forms spatially separated domains of proliferation and differentiation. The data presented in this manuscript potentially suggest that serum starvation works through adherens junctions to create differentially dense fields within the cultures which determines whether cells proliferate or differentiate. The authors then perform experiments to show that junction formation with starvation drive keratinocyte differentiation potentially through YAP signaling. However, these experiments are rather loosely connected and their results often do not support the conclusions drawn by the authors. However, the not well supported conclusion the form the basis for a fact statement, but their data really did not show that. For example, the authors state: "By contrast, YAP inhibition by a tankyrase inhibitor, XAV939, suppressed pattern-dependent proliferation (Fig. 4i, j)," . However, their data do not show that proliferation is pattern-dependent but is nevertheless used to connect to and draw a conclusion about YAP signaling. The data itself appear to be of high quality, figures are well organized and statistics of quantification seem appropriate, but it somewhat problematic that throughout the manuscript it remains unclear if certain statements are hypotheses or conclusions on real data. Pattern formation as a requirement for differentiation is an interesting concept. However, the presented study lacks proper conclusive data how these patterns may contribute to proliferation and differentiation and remains rather short on what exactly is the instructive nature of these patterns, as they only use high density and are not generating own patterns with defined cues that explore what cues contribute.

Major points:

The statement "According to the RNA-seq data, AJ molecules, such as E-cadherin and actin, were localized at intercellular junctions in areas of high cell density" is not correct. RNA-seq does not allow conclusions about protein localization. Instead, the GO-Term analysis shown in Figure 2b shows downregulation of "cell-adhesion" in dense areas.

According to the reviewer's suggestion, we have corrected the sentence. We have reanalyzed our RNA-seq data and confirmed that GO-term "cell-adhesion" was in the top list of both high and low cell density regions. We have provided more data to the revised manuscript (Supplementary Fig. 7 and Supplementary Table 1).

Consistently the E-cadherin staining presented in Figure2c suggest lower intercellular E-cadherin levels in the most dense areas. However, any statement about junctional localization of adhesion components requires e.g. intensity quantification at junctions vs. cytoplasm or else to discriminate from intense overall staining due to high cell density and thus high overall junction numbers.

Actually, junctional E-cadherin was more pronounced in the high cell density area. We have provided line plot data to confirm this and also added a quantification data (Supplementary Fig. 8).

Hence, even though potentially true, the statement: "These data suggest that cells in regions of high cell density form AJs in response to intercellular forces" is not fully supported by the data shown so far.

Please see the answer to the comments of Reviewer 1 on the quantification of α 18. We believe that, as α 18 intensity is more pronounced in the cell-cell junction of high cell density area compared with low cell density regions, our claim is experimentally supported.

The authors suggest a pattern of high and low density that is formed over time. However, at the same time high density areas show formation of a second layer. Hence, "denser" areas as observed by phase contrast images or DAPI positive nuclei may either represent dense or stratified cells. What is missing is an analysis of cell density before cells started to stratify making sure only cells in the basal layer are analyzed. Otherwise, density and stratification which are perhaps interdependent in this system cannot be discriminated.

As the reviewer pointed out, the patterning was analyzed at the level of basal layer. In addition to Figure 1c, we have provided another plane cut immunofluorescence data (Supplementary Fig. 2) to the revised manuscript to address this issue.

The mathematical model does not include stratification and it is thus not clear to what extent it may explain the observed patterns.

It is true that the model does not account for stratification. It focuses solely on the patterning of cell density in the basal layer. We have incorporated this notion into the revised manuscript as a limitation of this study.

Moreover, the model appears to assume variables that have not been determined or cited. This reviewer is not an expert in modeling and thus cannot fully judge the math behind the model. However, the model appears to be biased if it assumes, as mentioned, that cell-adhesion increases with density.

The first equation, describing the time evolution of ρ (the cell density), incorporates diffusion, collective cell movement due to stress from adjacent cells, and random fluctuations. Each of these terms comes from a general consideration of density dynamics. The second equation, describing stress balance, is a generalization of Reference 72 (originally Reference 51). The crucial assumption here is that the cell-cell adhesion increases with density, which corresponds to the experimental findings (Revised Fig. 3a, b, Supplementary Figure 8a-h).

What we have demonstrated here is that we only need cell-cell adhesion as a source of attractive interactions for cells to form the density patterning as observed in the experiment. Since it is not self-evident whether the assumption of the density-dependent adhesion entails the emergence of density patterns, we do not believe that our model is begging the question or biased.

If low adhesion forces do not produce patterns, what is the counterforce in the model? Are cells allowed to change size to enable low density areas or do cells lose contact with neighbors despite high adhesion strength?

Our model does not have a variable corresponding to cell-shape change, which is considered only implicitly: Cells in the low density region (small ρ) are regarded as flattened, whereas those in the high density region (large ρ) as compressed (though not stratified) (Fig 1c).

The behavior of the model is controlled by the parameter β : a smaller β means that density variations have little effect on stress, whereas a larger β leads to significant stress changes with

density variation. Since stress increases as $\beta \cdot \rho$ (in the second equation), stress in the low density region remains low even when the parameter β is large.

Overall, it appears that the model is set up such, that it tends to reproduce what was observed in experiment. This conclusion, however, may result of an incomplete understanding of the model parameters.

The model setup, the assumption on the relationship between density and cell-cell adhesion in particular, does not inherently dictate the emergence of high/low density patterns: It might be the case that cell density is uniformly distributed everywhere with uniformly strong adhesion among cells. What our computer simulations have shown, however, is that the model exhibits spatially heterogeneous density patterns for sufficiently high β values. The emergence of such spatial patterns is not a predefined aspect of the mathematical model itself.

In the revised manuscript, the non-triviality of the spatial patterning has been made clear in the Results, and more explanations on the mathematical model to address the above points have been added to the Methods section.

If dense areas do actually represent stratified areas it may not be surprising that the GO analysis indicates an increase in differentiation. A requirement for AJ or intercellular junctions in general is less surprising as stratification requires cell-cell adhesion. The observation that AJ are essential for intercellular junction formation in keratinocytes or in other epithelial cells is not new (e.g. Michels et al. JID 2009).

We agree with the reviewer in the point that the role of AJ is not new. We have incorporated the notion into the Discussion of the revised manuscript and cited the paper the reviewer indicated.

The part of the paper addressing the role of YAP suffers from a number of potentially mislead assumptions/conclusions based on a previous experiment which then did not properly supported that conclusion (see also overall comments). For example, the statement "YAP inhibition by a tankyrase inhibitor, XAV939, suppressed pattern-dependent proliferation" contains interdependencies that have not been show [sic]. XAV939 may just inhibit proliferation which is not necessarily pattern dependent. Too much speculation confuses data and hypotheses.

We agree with the reviewer to point out that pattern-dependency was not supported by our results. We have reduced the tones and corrected these terms in the revised manuscript.

The 3D HaCaT cultures are performed on transwell filters with medium supply above and below cells, with the assumption that organizing patterns are also formed under these conditions. However, this has not been shown by the authors. Their suggestion that serum starvation may increases thickness of cultures through alterations in the organization of [sic]

We showed the patterning in air-liquid interface culture in the Supplementary Data (Supplementary Fig. 12a, b, original Supplementary Fig. 9a, b), which presents starvation-induced patterning even in such condition.

Reviewer #3 (Significance (Required)):

The mechanisms that drive self-organization of epithelial cells to spatially separate domains of proliferation and differentiation is in principle a very interesting topic of high interest to the cell and mechanobiology community, [sic]

We appreciate the reviewer's constructive and thoughtful comments.

June 24, 2024

RE: Life Science Alliance Manuscript #LSA-2024-02893

Dr. Ken Natsuga
Hokkaido University Graduate School of Medicine
Department of Dermatology
Sapporo 060-8638
JAPAN

Dear Dr. Natsuga,

Thank you for submitting your revised manuscript entitled "Cell-cell adhesion drives patterning in stratified epithelia". We would be happy to publish your paper in Life Science Alliance pending final revisions necessary to meet our formatting guidelines.

- please be sure that the authorship listing and order is correct
- please add your main, supplementary figure, table, and movie legends to the main manuscript text after the references section
- please add a Summary Blurb/Alternate Abstract and a Category to our system
- please add the Twitter handle of your host institute/organization as well as your own or/and one of the authors in our system
- title in the system and manuscript file must match
- please add an Author Contributions to our system as well
- since Figure 8 is a Graphical Abstract, please upload it with that file designation and remove the legend for it
- please remove figures from the manuscript file; they should be uploaded only separately
- there is a callout for Figure 8A-C; should this be S8A-C? Please check
- please add callouts for Figure S9A-B to your main manuscript text

A. FINAL FILES:

B. MANUSCRIPT ORGANIZATION AND FORMATTING:

Thank you for your attention to these final processing requirements. Please revise and format the manuscript and upload materials within 5 days.

Sincerely,

June 26, 2024

RE: Life Science Alliance Manuscript #LSA-2024-02893R

Dr. Ken Natsuga
Hokkaido University
Department of Dermatology
N15 W7
Kita-ku
Sapporo, Hokkaido 060-8638
Japan

Dear Dr. Natsuga,

Thank you for submitting your Research Article entitled "Patterning in stratified epithelia depends on cell-cell adhesion". It is a pleasure to let you know that your manuscript is now accepted for publication in Life Science Alliance. Congratulations on this interesting work.

DISTRIBUTION OF MATERIALS:

Again, congratulations on a very nice paper. I hope you found the review process to be constructive and are pleased with how the manuscript was handled editorially. We look forward to future exciting submissions from your lab.

Sincerely,
